# Comprehensive targeting of resistance to inhibition of RTK signaling pathways by using glucocorticoids

Ke Gong[1,2], Gao Guo[1], Nicole A. Beckley[1], Xiaoyao Yang[1], Yue Zhang[1], David E. Gerber [3,4,5], John D. Minna [6,7], Sandeep Burma [8,9], Dawen Zhao [10], Esra A. Akbay [11] & Amyn A. Habib [1,4,12 ✉]

Inhibition of RTK pathways in cancer triggers an adaptive response that promotes therapeutic resistance. Because the adaptive response is multifaceted, the optimal approach to blunting it remains undetermined. TNF upregulation is a biologically significant response to EGFR inhibition in NSCLC. Here, we compared a specific TNF inhibitor (etanercept) to thalidomide and prednisone, two drugs that block TNF and also other inflammatory pathways. Prednisone is significantly more effective in suppressing EGFR inhibition-induced inflammatory signals. Remarkably, prednisone induces a shutdown of bypass RTK signaling and inhibits key resistance signals such as STAT3, YAP and TNF-NF-κB. Combined with EGFR inhibition, prednisone is significantly superior to etanercept or thalidomide in durably suppressing tumor growth in multiple mouse models, indicating that a broad suppression of adaptive signals is more effective than blocking a single component. We identify prednisone as a drug that can effectively inhibit adaptive resistance with acceptable toxicity in NSCLC and other cancers.

[1] Department of Neurology, University of Texas Southwestern Medical Center, Dallas, TX 75390, USA. [2] Hubei Province Key Laboratory of Allergy and Immunology and Department of Immunology, School of Basic Medical Sciences, Wuhan University, Wuhan 430071, China. [3] Division of Hematology-Oncology, Department of Internal Medicine, University of Texas Southwestern Medical Center, Dallas, TX 75390, USA. [4] Harold C. Simmons Comprehensive Cancer Center, University of Texas Southwestern Medical Center, Dallas, TX 75390, USA. [5] Department of Population and Data Sciences, University of Texas Southwestern Medical Center, Dallas, TX 75390, USA. [6] Department of Pharmacology, University of Texas Southwestern Medical Center, Dallas, TX 75390, USA. [7] Hamon Center for Therapeutic Oncology Research, University of Texas Southwestern Medical Center, Dallas, TX 75390, USA. [8] Department of Neurosurgery, University of Texas Health San Antonio, San Antonio, TX 78229, USA. [9] Biochemistry & Structural Biology, University of Texas Health San Antonio, San Antonio, TX 78229, USA. [10] Departments of Biomedical Engineering, Wake Forest School of Medicine, Winston Salem, NC 27157, USA. [11] Department of Pathology, University of Texas Southwestern Medical Center, Dallas, TX 75390, USA. [12] VA North Texas Health Care System, Dallas, TX 75216, USA. ✉email: Amyn.Habib@UTSouthwestern.edu

Resistance to targeted treatment is a major problem in the treatment of cancer. The EGFR is widely expressed in non-small cell lung cancer (NSCLC) and appeared to be a promising target in NSCLC[1–4]. However, EGFR inhibition using tyrosine kinase inhibitors (TKIs) is highly effective initially, but only in the minority of patients with EGFR-activating mutations[5–7]. EGFR expression is detected in up to 90% of NSCLC and most lung cancers express EGFR wild type (EGFRwt)[1–4]. The presence of EGFR ligands is common in lung cancer[8,9] and a constitutive overexpression-induced EGFRwt signaling has also been reported[10–12] suggesting that EGFRwt is activated and likely plays an oncogenic role in lung cancer[8]. Nonetheless, EGFRwt tumors exhibit a primary resistance and do not respond to EGFR inhibition. Moreover, patients with EGFR mutant NSCLC who clearly respond to EGFR inhibition inevitably develop a secondary resistance[3]. The emergence of secondary resistance implies the persistence of subsets of cancer cells that are not eliminated during the initial treatment and is a major therapeutic hurdle. The mechanisms of resistance to EGFR inhibition in NSCLC is an intensively researched area[3,13,14]. Both mutational and non-mutational mechanisms of resistance have been described[15]. For example, major mutational mechanisms of resistance to EGFR inhibition in NSCLC include secondary EGFR mutations such as the T790M mutation[16] and amplification of other RTKs such as MET[17]. These genetic changes are detected after months of exposure to TKIs.

In addition, inhibition of RTK signaling pathways in cancer cells leads to a rapid reprogramming of signaling pathways that frequently leads to a resumption of previously suppressed signals or activation of alternative signals that are functionally similar[15,18–22]. This adaptive response may protect cells from a loss of RTK signals[3] and may play an important role in mediating both primary and secondary resistance[23]. Since EGFR mutant NSCLC patients respond to EGFR TKIs clinically, adaptive responses do not seem to impact the initial response to EGFR inhibition, although they may play a role in secondary resistance. Resistance to EGFR inhibition in NSCLC has been studied primarily in EGFR mutant tumors[3,15]. However, recent studies indicate that inhibition of EGFRwt also triggers a robust adaptive response. This adaptive response is broad and may involve multiple bypass signaling pathways such as STAT3 or NF-κB[15,24,25]. A significant component of this adaptive response includes activation of inflammatory signaling pathways[4,24,26,27]. A previous study of RAF inhibition using vemurafenib in melanoma revealed activation of at least 6 signaling pathways[28,29]. We have recently reported that a rapid TNF-driven resistance mechanism plays a key role in resistance to EGFR inhibition in NSCLC[26] and in glioma[30–32].

Glucocorticoids (GCs) have been used extensively for a variety of inflammatory and other conditions for decades[29,33]. GCs act by binding to the glucocorticoid receptor (GCR) which subsequently translocates to the nucleus and acts as a transcriptional activator or repressor or acts via protein-protein interactions with other transcription factors[34]. GCs are potent suppressors of TNF-NF-κB signaling[29,35], inhibit a broad range of cytokines, and regulate multiple facets of the inflammatory and immune response[33]. Etanercept (Enbrel) is a fusion protein of TNFR and IgG1 and is in clinical use as a stable and effective TNF blocking agent for rheumatologic diseases[36]. Thalidomide is a known inhibitor of TNF and may regulate TNF transcription and/or stability[36,37], but also influences multiple additional targets.

There is increasing evidence that long-term efficacy of treatment targeted at RTK signaling pathways will require suppression of the accompanying adaptive resistance[38]. Thus, an early or delayed adaptive response may help subpopulations of cancer cells to survive targeted treatment and eventually lead to secondary resistance. The adaptive response may also play an important role in primary resistance. Since the adaptive response is broad and multipronged, one would expect a comprehensive targeting of multiple signals to be optimal[25]. Alternatively, although EGFR inhibition leads to altered expression of a large number of genes, a small number of important signaling pathways could mediate resistance, and inhibition of a single key pathway could cripple the adaptive response and be sufficient to overcome therapeutic resistance to EGFR inhibition. In this study we examine the efficacy of a broad vs. focused inhibition of the adaptive response in NSCLC using TNF as a representative component of the adaptive response.

## Results

**The transcriptional response to EGFR inhibition in NSCLC.** Our recent studies have focused on individual components of the adaptive resistance to EGFR inhibition in cancer[26,27,30–32]. In order to better understand the scope of the early adaptive response to EGFR inhibition, we recently undertook an unbiased transcriptome analysis in the EGFRwt/KRAS mutant A549 cell line after treatment with erlotinib. Our data indicated that EGFR inhibition leads to a broad transcriptional response and we identified a large number of erlotinib-upregulated genes by RNAseq[27]. In another study, we had reported that TNF upregulation is a universal and biologically significant adaptive response to EGFR inhibition in EGFR mutant and EGFRwt expressing NSCLC[26]. In that study, we had used etanercept and thalidomide as TNF blockers[26]. To examine the contribution of TNF signaling to this adaptive response to EGFR inhibition, we repeated our RNAseq experiment (Fig. 1a) and inhibited TNF in addition to the EGFR in our RNAseq analysis by using etanercept, thalidomide, or prednisolone. Prednisone is a widely used drug in clinical practice as a broad inhibitor of inflammation that also inhibits TNF and the transcription factor NF-κB[29,35]. Prednisone is a synthetic GC prodrug that is converted by the liver into prednisolone (a beta-hydroxy group instead of the oxo group at position 11), which is the active steroid. In this study, prednisolone was used in in vitro studies, while prednisone was used in mouse experiments. The abbreviation PRDN represents both prednisone and prednisolone, since they both work finally in the form of prednisolone. We examined the efficacy of each of these drugs to inhibit the transcriptional response to EGFR inhibition. Overall, we found that etanercept, a specific TNF blocker, inhibited about 31% of the erlotinib-induced gene upregulation while thalidomide blocked about 42%. Prednisolone had the most robust effect and inhibited about 66% of genes upregulated by EGFR inhibition (Fig. 1b). We also found that multiple key signal transduction pathways are also induced by EGFR inhibition and are most efficiently blocked by prednisolone (Supplementary Fig. 1A).

**Suppression of TNF-NF-κB and inflammatory cytokines in response to EGFR inhibition.** Although erlotinib-induced genes in the TNF pathway were blocked by the three drugs at different strength, the TNF gene were almost equally blocked by the three inhibitors (Fig. 1c). We confirmed that etanercept, thalidomide, and prednisolone inhibited the erlotinib-induced upregulation of TNF in multiple NSCLC cell lines expressing EGFRwt or EGFR mutant at an mRNA and protein level (Fig. 1d–g, Supplementary Fig. 1B–I). In addition, etanercept, thalidomide, and prednisolone all inhibited TNF-induced activation of NF-κB as determined by an NF-κB reporter assay and a block of IKBα degradation in response to TNF (Fig. 1h–o). We found that prednisolone uses multiple mechanisms to suppress activation of the key survival transcription factor NF-κB in response to erlotinib. Prednisolone regulates both the canonical and non-canonical NF-κB pathways via distinct

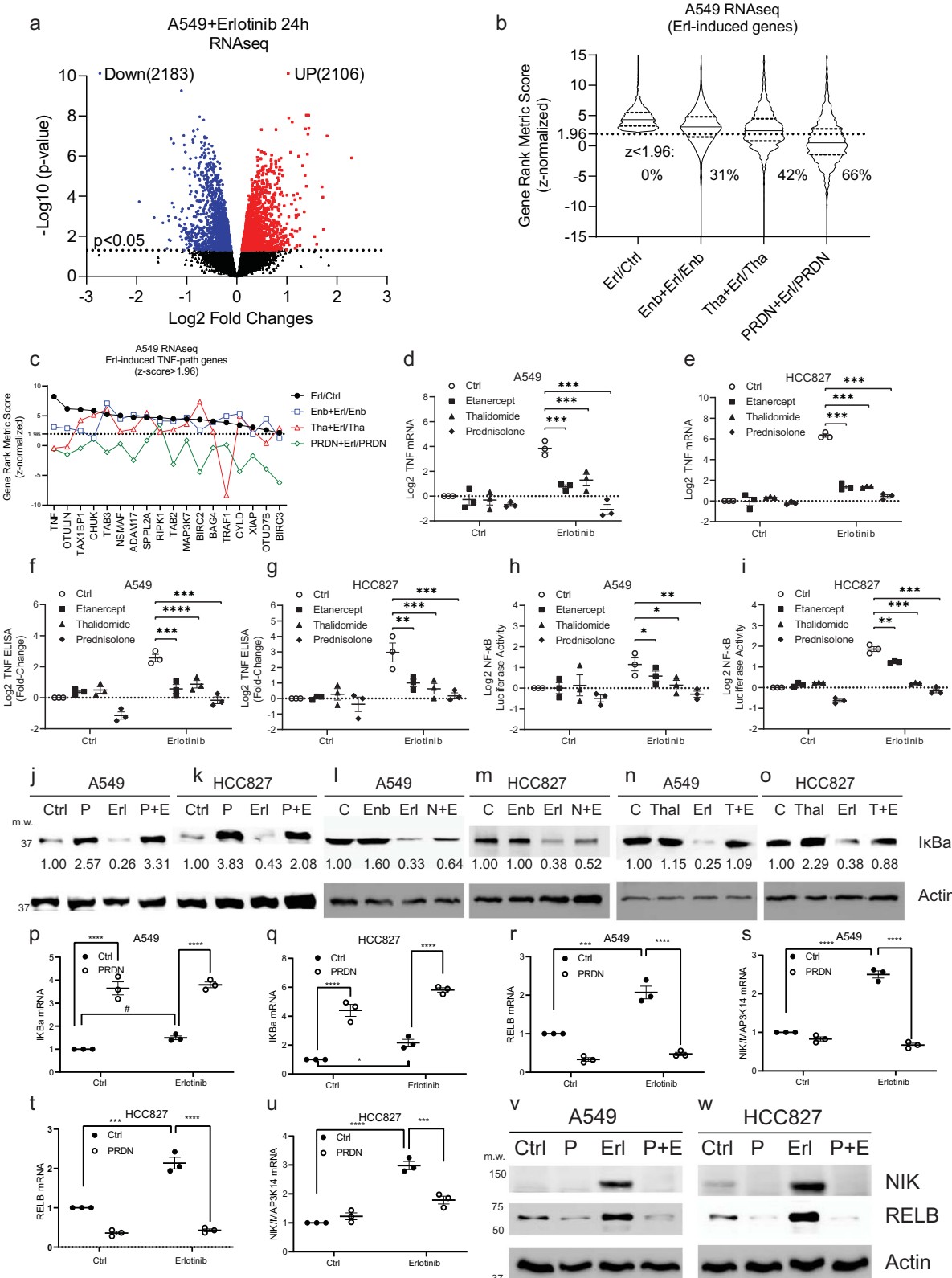

mechanisms. Consistent with previous work[39], we also found that prednisolone induces upregulation of IKBα, an inhibitor of NF-κB as the mechanism for inhibiting the canonical pathway for activation of NF-κB in NSCLC cell lines (Fig. 1j, k, p–q). To inhibit the non-canonical activation of NF-κB, prednisolone downregulates NF-κB subunits such as RELB, and other proteins related to NF-κB activation such as MAP3K14 (NF-κB-inducing kinase, NIK) in

both EGFRwt and EGFR mutant NSCLC cell lines at an mRNA and protein level (Fig. 1r–w).

TNF is not the only cytokine upregulated by EGFR inhibition. RNA-seq analysis revealed that multiple inflammatory cytokines are upregulated in response to EGFR inhibition. Importantly, concomitant use of prednisolone resulted in a broad suppression of erlotinib-induced cytokines (Supplementary Table 1A). Both

**Fig. 1 Erlotinib-induced gene expression in A549 cells is differentially suppressed by various TNF inhibitors. a** EGFRwt lung cancer adenocarcinoma line A549 was treated for 24 h with vehicle or 1 μM erlotinib, with/without 100 μg/mL Enbrel (Enb/etanercept), 10 μM thalidomide (Tha), 10 μM prednisolone (PRDN). RNAseq was performed on 24 RNA samples in three independent experiments as in the Methods. Volcano plot shows the differentially expressed genes affected by erlotinib, identified by EdgeR. **b** Violin plot shows the genes with Gene Set Enrichment Analysis (GSEA) gene ranking metric z-normalized scores in the comparison of erlotinib and control > 1.96 ($p < 0.05$) and false discovery rate (FDR) in the Benjamini-Hochberg method < 0.05. These erlotinib-upregulated genes would have 31%, 42%, and 66% blocked (z-score < 1.96) by Enbrel, thalidomide, and prednisolone, respectively. **c** Among genes in the TNF signaling pathway (Reactome Pathway Database), erlotinib-induced 18 leading-edge genes with z-score > 1.96 were blocked by the 3 TNF inhibitors at different strengths, via comparing erlotinib-induced GSEA metric scores between absence and presence of each inhibitor. **d, e** Real-time PCR was performed to validate the erlotinib-induced TNF mRNA can be blocked by these three drugs in EGFRwt and EGFR mutant lung cancer cells. **f, g** TNF protein levels were examined by Enzyme-Linked Immunosorbent Assay (ELISA). **h, i** Erlotinib-increased NF-κB activity was blocked by these three drugs in luciferase reporter assays. **j–o** Western Blot (WB) shows the IκBa degradation was blocked by these three drugs. Specifically, prednisolone was able to increase IκBa expression. Data were quantified by ImageJ. **p–q** Prednisolone-increased IκBa mRNA levels were determined by real-time PCR. **r–w** Non-canonical NF-κB pathway components NIK/MAP3K14 and RELB was also induced by erlotinib and blocked by prednisolone in mRNA and protein levels. 100 nM erlotinib is used for HCC827. Treatments last for 6 (**j–q**) or 24 h (others). Real-time PCR were performed in 3 independent experiments, showing each value and mean ± Standard Error of the Mean (SEM). WB images are one representative from at least three independent experiments. *$p < 0.05$, **$p < 0.01$, ***$p < 0.001$, ****$p < 0.0001$, by two-way analysis of variance (ANOVA), with Bonferroni-adjusted. The statistical analysis above was performed on Graphpad Prism 9.0.0. Source data are provided as a Source Data file. $p = 0.0001, 0.001, 2E−6; 8E−7, 3E−10, 1E−9; 1E−4, 6E−4, 2E−5; 0.006, 0.001, 0.001; 0.04, 0.03, 0.01; 0.002, 1E−7, 3E−7$ (**d–i**). $5E−6, 1E−5; 2E−5, 1E−5; 5E−5, 3E−6; 2E−7, 3E−8; 1E−5, 2E−6; 4E−6, 1E−7$ (**p–u**).

etanercept and thalidomide also suppress cytokine induction in response to EGFR inhibition but to a lesser degree than prednisolone.

**Inhibition of receptor tyrosine kinase activation**. Activation of bypass RTK signaling pathways is a central component of the early adaptive response to EGFR inhibition in cancer cells. Alternative RTK signaling pathways may be activated by inhibition of a specific receptor, for example, EGFR inhibition leading to Axl activation. Alternatively, the basal activity of other RTKs not induced by EGFR inhibition may also play a role in resistance. A particularly important role for RTK ligands has been reported in mediating secondary resistance in NSCLC[40,41]. Biologically, bypass RTK signaling pathways play a compensatory role to protect cancer cells from a loss of RTK signaling and are likely to play a key role in primary resistance. In oncogene addicted cells activation of RTK bypass signaling triggered by EGFR (or other RTK) inhibition could promote survival of subsets of tumor cells that lead eventually to secondary resistance. Our transcriptome analysis indicated that erlotinib upregulates ligands for multiple RTKs, and that this upregulation is blocked by prednisolone but not by etanercept (Enbrel) or thalidomide (Supplementary Table 2). We confirmed the erlotinib-induced upregulation of a broad range of RTK ligands and a suppression of erlotinib-induced upregulation of RTK ligands by prednisolone using qPCR in multiple EGFRwt and EGFR mutant cell lines (Fig. 2a, d, Supplementary Table 1B, C, and Supplementary Fig. 2A, B). We confirmed the qPCR results by conducting ELISA for HGF and heregulin (Fig. 2b, c, e, f and Supplementary Fig. 2C–F). These data indicate an almost total suppression of erlotinib-induced RTK ligands by prednisolone. Next, we examined the activation of RTKs in NSCLC lines exposed to erlotinib by using human phospho-RTK array kit that detects the activation of multiple RTKs on a single membrane. We found that EGFR inhibition frequently led to an initial decrease in activation of multiple RTKs, suggesting that the EGFR is partially driving the activation of multiple other RTKs. However, as part of the adaptive response, these RTKs become reactivated after 24–72 h of EGFR inhibition, even though the EGFR kinase activity remains suppressed. Importantly, this reactivation is almost completely suppressed by prednisolone in both EGFRwt and EGFR mutant NSCLC cell lines (Fig. 3a, b and Supplementary Fig. 3A, B). We compared the effect of prednisolone in suppressing RTK activation to etanercept and thalidomide and found that prednisolone is significantly more potent in suppressing

adaptive or basal RTK activation compared to either etanercept or thalidomide. (Supplementary Fig. 4). Prednisolone also suppressed RTK activation induced by use of the third-generation EGFR TKI osimertinib in EGFR mutant cell lines (Supplementary Fig. 5). The prednisolone-mediated suppression of bypass RTK signaling in cancer cells is likely a key factor in the highly effective synergism detected between EGFR inhibition and prednisone in animal models discussed below. The RTKs upregulated by erlotinib and suppressed by prednisolone include MET, ERBB2, ERBB3, and RET among others. Prednisolone also inhibits the basal activity of RTKs not activated by erlotinib, resulting in a broad suppression of multiple RTKs. There are some exceptions to the suppressive effect of prednisolone on RTKs. Consistent with its role as a GC, prednisolone tends to activate insulin receptor (IR) or insulin-like growth factor receptors (IGF1R) (Fig. 3a, b and Supplementary Fig. 3A, B). Next we confirmed the results of our phospho-kinase array results by performing Western blots for representative targets. First, we undertook a time course of activation of key RTKs and downstream signals in response to erlotinib in EGFRwt and EGFR mutant cell lines (Fig. 3c, d). We examined the activation of the receptors MET, ERBB2, ERBB3, and RET and found that adaptive reactivation of all these receptors was blocked by prednisolone (Fig. 3e, f). Next, we examined RTK driven downstream signals such as STAT3 and YAP that are known to be activated by EGFR inhibition in different cell lines and to play a key role in the adaptive response and in mediating therapeutic resistance to EGFR inhibition[15,38,42–46]. While the adaptive signals triggered by EGFR inhibition vary among cell lines we found that prednisolone completely suppressed the adaptive activation of STAT3 and YAP (Fig. 3g–i). There is some variation in the adaptive responses triggered in different cell lines by EGFR inhibition and we did not detect an activation of STAT3 in response to EGFR inhibition in A549, HCC827, or H441 cells (Fig. 3e, f, i)[15]. Thus, prednisolone is more effective than etanercept or thalidomide in suppressing a broad range of adaptive resistance signals triggered by EGFR inhibition in multiple EGFR mutant and EGFRwt NSCLC lines. Similar results were found with dexamethasone (Supplementary Fig. 6). Erlotinib-induced STAT3 activation was previously reported to be mediated via IL-6/Jak1 and we also found that IL-6 mRNA levels are suppressed by prednisolone (Supplementary Fig. 7).

**A broad suppression of transcription induced by prednisolone**. Prednisone is known to act as a broad transcriptional repressor[47].

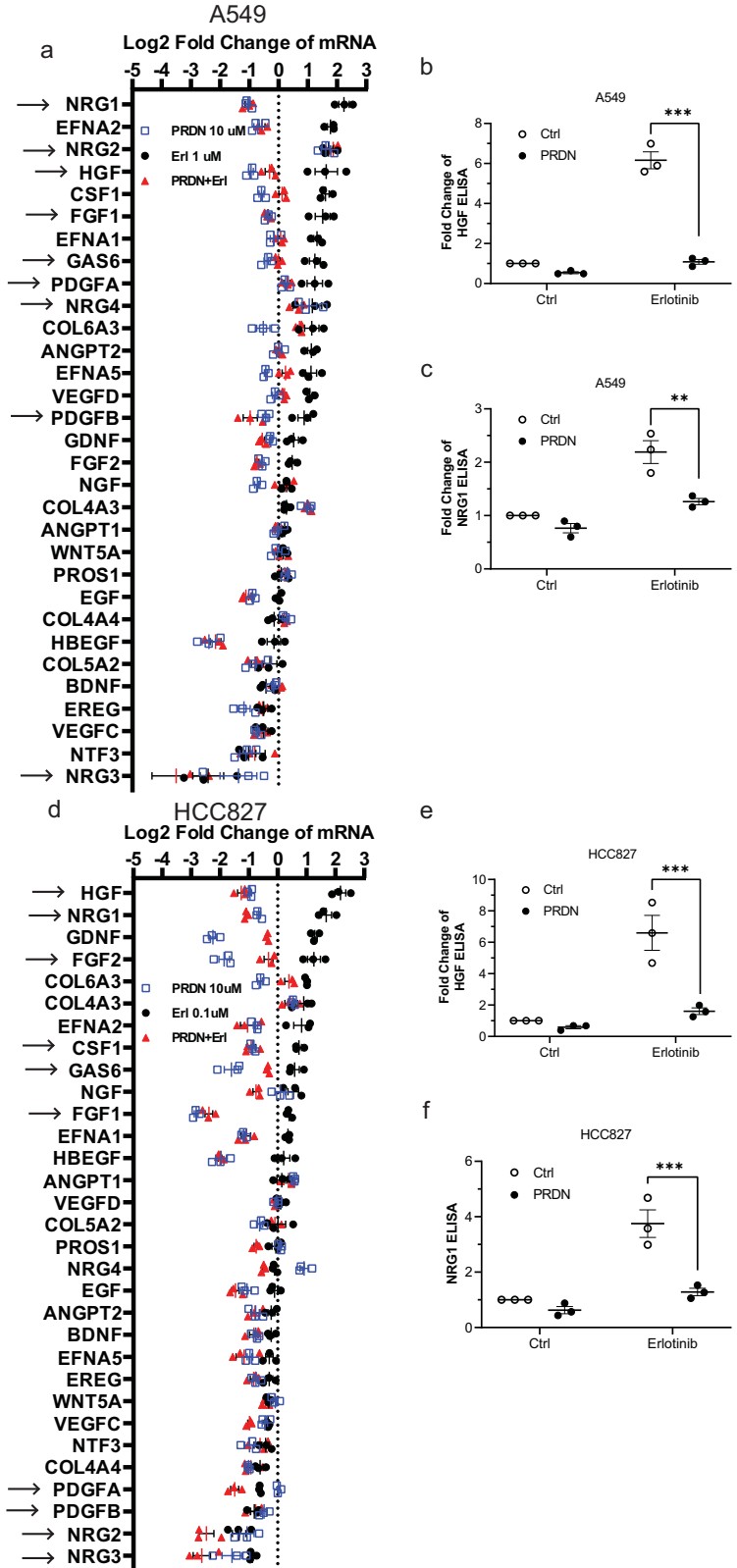

EGFR inhibition is known to result in activation of multiple transcription factors[15]. We examined the effect of prednisolone on the activation of transcription factors induced by erlotinib in HCC827 and A549 cells using the CIGNAL 45-Pathway Reporter Array that measures activation of forty-five pathways by determining the activities of their transcription factors. As expected, we found that EGFR inhibition induced the activation of multiple transcription factors in A549 cells. Prednisolone suppressed the basal activation of these transcription factors and neutralized the erlotinib-induced upregulation of these transcription factors, while increasing activation of a few (Supplementary Fig. 8A–D). A similar result was found in HCC827 cells, although the number of transcription factors induced by erlotinib in these cells is higher (Supplementary Fig. 8E–H). These data suggest that the

**Fig. 2 Erlotinib upregulates a large number of RTK ligands and most were suppressed by concomitant prednisolone. a** EGFRwt A549 cells were treated with 1 μM erlotinib with or without 10 μM prednisolone for 3 days. Total RNA was extracted and subjected to real-time PCR for detecting mRNA levels of multiple RTK ligands as listed. Real-time PCR were performed in three independent experiments, showing each value and mean ± SEM. The mean values, p-values, and q-values of multiple t-tests were summarized in Supplementary Table 1. The multiple comparison correction was corrected by two-stage step-up method of Benjamini, Krieger and Yekutieli. The desired false discovery rate (FDR) was set as 0.05. **b, c** HGF and NRG1 protein levels were validated by ELISA, under the same treatment condition as PCR. Three independent experiments show each result and mean ± SEM. **p < 0.01, ***p < 0.001 by two-way analysis of variance (ANOVA), adjusted by Bonferroni's test. **d–f** Similar experiments were performed on EGFR mutant HCC827 cells, except that the dose of erlotinib was 100 nM. The statistical analysis above was performed on Graphpad Prism 9.0.0. Source data are provided as a Source Data file. p = 3E−7,0.001;0.0005,4E−4.

major mechanism used by prednisolone to block the erlotinib-induced response is transcriptional repression.

**The effects of GCs and EGFR inhibition on the glucocorticoid receptor (GR).** The effects of GCs are mediated by the GR. We first examined whether the level of GR is altered in response to either erlotinib or GCs or a combination. GR levels were examined following treatment of lung cancer cell lines with EGFR inhibitor, prednisolone or a combination for 24 h (Supplementary Fig. 9A). We did not detect a significant change in GR level. As expected, we see a substantial increase in phosphorylation of the GR in response to GCs, but not to erlotinib (Supplementary Fig. 9A). We next examined GR levels after a 7-day exposure to these drugs. Although a significant downregulation of the GR upon GCs treatment was observed, GR was still presented (Supplementary Fig. 9B). Similarly, we also examined GR levels in resected tumor tissue after one month of treatment with GC, erlotinib or a combination and found GR was partially reduced by GC treatment (Supplementary Fig. 9C). We did not find a significant effect of erlotinib on GR levels. We also examined the transcriptional activity of the GR using a luciferase reporter. As expected prednisolone induced a strong response, while erlotinib alone has no effect and the combination of prednisolone and erlotinib is similar to prednisolone alone (Supplementary Fig. 9D, E).

**Prednisolone enhances sensitivity to EGFR inhibition.** Next, we examined whether prednisolone enhances sensitivity to EGFR inhibition using cell survival assays. We found that all 3 TNF blocking drugs etanercept, thalidomide, or prednisolone rendered EGFRwt/KRas mutant expressing A549 cells sensitive to EGFR inhibition (Supplementary Fig. 10A). We found similar results with two additional EGFRwt expressing cell lines that are normally resistant to EGFR inhibition (Supplementary Fig. 10B, C). Next we examined the effect of combining TNF and EGFR inhibition in lung cancer cell lines with EGFR activating mutations. A similar result was obtained in HCC827, H3255, and PC9 cells with a combination of erlotinib and one of the TNF inhibiting drugs providing synergistic suppression of viability (Supplementary Fig. 10D–F). We also tested afatinib, a second-generation EGFR inhibitor, and found similar results (Supplementary Fig. 10G, H). We also compared the effect of a combination of erlotinib plus prednisolone in colony formation assays and found a more potent suppression with erlotinib and prednisolone in multiple cell lines compared to erlotinib plus etanercept or thalidomide (Supplementary Fig. 10I-K).

**Biological effects of EGFR inhibition plus prednisone in vivo.** Next, we examined whether a combined inhibition of TNF and EGFR would influence sensitivity to erlotinib in mouse xenograft models. We started our studies with the EGFRwt/KRas mutant/STK11 mutant A549 cells known to be highly resistant to EGFR inhibition. We tested the efficacy of the EGFR TKI erlotinib in

combination with various TNF inhibitors. A549 cells were injected into the flanks of nude mice to form subcutaneous tumors. Once tumors became visible, treatment was started with control vehicle, etanercept, thalidomide, or prednisone either alone or in combination with erlotinib. As expected, we found robust tumor growth in controls. The erlotinib, etanercept, thalidomide, prednisone alone treated groups did not have a significant effect on tumor growth. A combination of erlotinib plus etanercept or a combination of erlotinib plus thalidomide had a significant effect on growth suppression as reported previously[26]. However, combined use of erlotinib plus prednisone had the strongest and most durable effect on growth suppression with a significant difference between erlotinib plus etanercept or thalidomide groups and erlotinib plus prednisone group (Fig. 4a, b). Thus, while there was a decrease in the effectiveness of the erlotinib plus etanercept and erlotinib plus thalidomide groups over time, in the erlotinib plus prednisone group tumor growth remained suppressed for the duration of the experiment which was 6 months (Fig. 4b). We conducted additional experiments and found that a combination of erlotinib plus prednisone is also effective if the combination treatment is started after tumors have grown to a large size as 1000 mm³ (Fig. 4c), or if prednisone is added after prior treatment of afatinib alone was given until tumors unaffectedly grow to around 1500 mm³ (Fig. 4d). In addition, when used in combination with EGFR inhibition, it is possible to taper the dose of prednisone after initially starting at a higher dose without a loss of tumor inhibition (Fig. 4e). Next, we examined the effect of erlotinib plus prednisone in a Patient-Derived Xenograft (PDX) model of NSCLC with KRAS G13C mutant and without activating EGFR mutations (HCC4087). We found the combination of afatinib plus prednisone to be highly effective in inhibiting the growth of this PDX tumor for an extended period of time (Fig. 4f). A similar result was obtained in HCC4087 xenografts with erlotinib plus prednisone (Fig. 4g). The combination of afatinib and prednisone is also highly effective in an H441 EGFRwt/KRas mutant/STK11 wt xenograft model (Fig. 4h). The dose of prednisone used in our study is quite consistent with use in human patients for various diseases. We use 5 and 1 mg/kg in our mouse models that are equal to 0.4 and 0.08 mg/kg in human, considering an adult weight of 60 kg, the equivalent dose would be 24 and 5 mg/day. Importantly, the prednisone dose can be tapered in our experimental paradigm to 1mg/day without a loss of efficacy (Fig. 4e), just as it is done in human patients. The dose calculation of mouse to human was performed as has been previously described[48].

We examined levels of RTK ligands, HGF and NRG1, and found significant upregulation of these ligands in tumor tissue from erlotinib-treated mice in both A549 and HCC827 tumors (Supplementary Fig. 11A–D). Similarly, these ligands were also upregulated in PDX derived xenograft tumors with or without EGFR activating mutations (Supplementary Fig. 11E–H). We also found that prednisone suppressed erlotinib-induced upregulation of RTK ligands in xenograft tumors with or without EGFR activating mutations (Supplementary Fig. 11I–P).

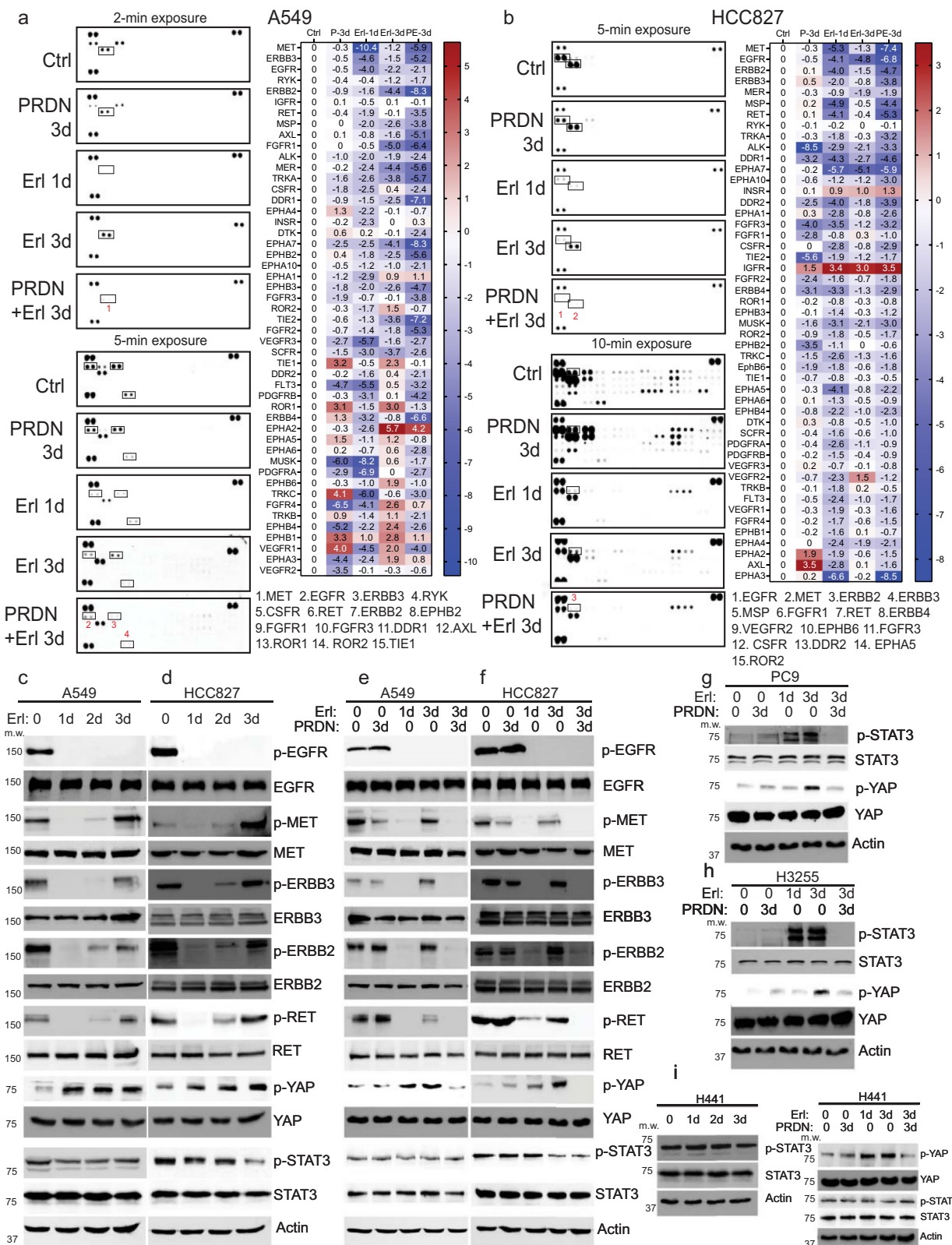

**Efficacy of erlotinib plus prednisone in immunocompetent models.** To further examine the effect of a combined EGFR plus TNF inhibition in an immunocompetent model, we used a well-established transgenic mouse model of lung cancer that is driven by doxycycline-mediated induction of the EGFR L858R mutation[49]. Once tumors were detected by magnetic resonance imaging (MRI), treatment was started with control vehicle, erlotinib, prednisone, or erlotinib plus prednisone as indicated in Fig. 5a. As expected, we found robust tumor growth in controls. Prednisone alone had no significant effect on tumor growth, while erlotinib alone suppressed tumor growth. However, the combination of erlotinib plus prednisone resulted in significantly more effective inhibition of tumor growth and animal survival than erlotinib alone, demonstrating that the combination

**Fig. 3 Bypass RTK signaling is suppressed by prednisolone in RTK arrays and Western blots. a** Proteome Profiler Human Phospho-RTK Array (R&D, ARY001B) was performed on 5 protein samples from A549 cells treated with control vehicle for 3 days, 10 μM prednisolone for 3 days, 1 μM erlotinib for 1 day and 3 days, and prednisolone + erlotinib for 3 days. Images were captured by ChemiDoc MP Imaging System (Bio-Rad) at exposures with varying time, processed by Image Lab 6.0.1 (Bio-Rad), and quantitated by HLImage++ (Western Vision Software). Some representative RTKs were framed up and footnoted. Additional images were shown in Supplementary Fig. 3. The quantitation was based on the longest exposure, except for the framed RTKs in shorter exposures to avoid overexposure. RTKs were ranked by basal expression. The log 2-fold changes of every treatment condition comparing to control were shown in the heatmap. **b** A similar experiment was performed on EGFR mutant HCC827 cells, where erlotinib was used at 100 nM. **c**, **d** EGFRwt A549 and mutant HCC827 cells were incubated with 1 μM and 100 nM erlotinib, respectively, for 0, 1, 2, and 3 days. Cell lysates were collected and detected by WB for total and phosphorylated EGFR, ERBB2, ERBB3, MET, RET, STAT3, and YAP. Actin was used as internal control. **e–i** EGFR mutant lines (HCC827, PC9, and H3255) and EGFRwt lines (A549 and H441) cells were treated with erlotinib or 10 μM prednisolone for the indicated days. The dosages for erlotinib are 1 μM for EGFRwt, 100 nM for HCC827, and 10 nM for PC9 and H3255. The indicated proteins were detected by WB. Each WB image is one representative from at least three independent experiments. Source data are provided as a Source Data file.

treatment is also effective in an immunocompetent model (Fig. 5a–c). Next, we used a well-established immunocompetent transgenic mouse model of KRAS mutant lung cancer driven by Adeno-CMV-Cre-mediated induction of KRAS G12D[50]. These tumors express EGFRwt and previous studies have shown that ERBB receptors play an important role in driving the growth of KRAS mutant tumors[51,52]. Once tumors were detected by MRI following adenovirus administration, treatment was started with control vehicle, erlotinib, prednisone, or erlotinib plus prednisone followed by periodic MRI imaging. While there is robust tumor growth in control and in the single drug groups, erlotinib plus prednisone was significantly effective in suppressing tumor growth and extending survival (Fig. 5d–f).

**A combination of EGFR inhibition plus prednisone confers sensitivity to EGFR mutant lung cancer cells with acquired resistance to EGFR TKI.** We have previously reported that HCC827 lines rendered experimentally resistant to EGFR inhibition[26] have elevated TNF levels and could be rendered sensitive to EGFR inhibition by concomitant use of TNF inhibition[26]. We examined whether a combination of EGFR TKI plus prednisone would render erlotinib-resistant HCC827 clones sensitive to EGFR inhibition. Indeed, we found that concomitant use of erlotinib plus prednisone rendered these cells sensitive to EGFR inhibition (Fig. 5g). In addition, as the T790M mutation is an important mechanism of secondary resistance and we tested H1975 cells (with EGFR L858R/T790M) using osimertinib and found that these cells also could be rendered sensitive to EGFR inhibition if prednisolone is used (Fig. 5h). Importantly, prednisone suppressed the bypass RTK activation in response to EGFR TKIs in both erlotinib-resistant HCC827 and H1975 cells (Supplementary Figs. 12–13). Next, we examined whether a combination of osimertinib plus prednisone would be effective in treating H1975 tumors in vivo. H1975 cells were injected into the flanks of mice to form subcutaneous tumors. Once tumors formed, treatment was started. Although osimertinib alone is effective in suppressing H1975 tumor growth, combined treatment with osimertinib and prednisone resulted in a further and highly effective suppression of tumor growth (Fig. 5i).

Next, we examined whether a combination of EGFR plus prednisone would prevent the development of the adaptive response in EGFR mutant models, which eventually leads to secondary resistance. HCC827 cells inoculated in mice subcutaneously and mice were treated with low dose erlotinib after the formation of tumors. After an initial period of slow growth, the tumors relapsed and started to grow again. However, if erlotinib was administered along with prednisone, we saw a sustained response on suppression of tumor growth (Fig. 5j). Thus, prednisone can overcome both intrinsic as well as acquired resistance to EGFR inhibition in multiple experimental models. In addition, a combination of erlotinib plus prednisone is more

effective compared to erlotinib plus thalidomide in an EGFR mutant PDX HCC4190 (Fig. 5k).

**Prednisone suppresses the adaptive response triggered in response to targeted inhibition of multiple RTK components pathways in various types of cancer.** Given the remarkable effect of prednisone on blocking the adaptive activation of RTK pathways in response to EGFR inhibition, we examined whether prednisone would also be effective in suppressing the adaptive resistance to non-EGFR RTK pathway targets. The H23 lung adenocarcinoma cell line is responsive to the MEK inhibitor trametinib. We found that the addition of trametinib led to bypass RTK activation in H23 cells and this RTK activation is significantly suppressed by prednisolone (Supplementary Fig. 14). We found that a combination of trametinib and prednisone was more effective than trametinib alone in a subcutaneous xenograft model (Fig. 6a). Similarly, in H1703 NSCLC cells with PDGFRA amplification that are moderately sensitive/resistant to imatinib, bypass RTK activation is suppressed by prednisolone (Supplementary Fig. 15), and prednisone plus imatinib synergistically block tumor growth (Fig. 6b). Next, we examined A2058 cells with a BRAF mutation however resistant to vemurafenib. We find that a combination of vemurafenib and prednisone is able to overcome the resistance of these cells to vemurafenib in a xenograft model (Fig. 6c). Similarly, bypass RTK activation in response to vemurafenib is suppressed by prednisolone in A2058 cells (Supplementary Fig. 16). We next examined gastric cancer-derived ERBB2 amplified OE19 cells that are sensitive to lapatinib and found that a combination of lapatinib plus prednisone is more effective than lapatinib alone in a xenograft model (Fig. 6d) and also detect a suppression of bypass RTK signaling by prednisolone (Supplementary Fig. 17). Finally, we examined the FGFR-TACC3 fusion, FGFR amplified bladder cancer RT112 cell line that is sensitive to BGJ398 (infigratinib) and found a similar synergy with prednisone in a xenograft model (Fig. 6e) and a suppression of bypass RTK signaling (Supplementary Fig. 18). Thus, the use of prednisone may have broad usefulness as an adjunct to RTK targeted treatment in multiple cancers.

**Evidence of bypass RTK signaling in NSCLC patient tissues.** We examined resected tumor tissue from TKI naïve and TKI treated patients and examined whether RTK ligands are upregulated in TKI treated tissue. Indeed, we found that HGF, NRG1, and GDNF levels are upregulated in TKI treated tissue compared to TKI naïve patients (Fig. 7a–c). Next, we generated three signatures of ligands upregulated by TKI treatment and suppressed by prednisolone, and non-RTK cytokines as well as RTK ligands (Fig. 7d–f). Finally, analysis of The Cancer Genome Atlas Program (TCGA) data reveals that the TKI-increased and GC-blocked ligands signature can predict an adverse prognosis in

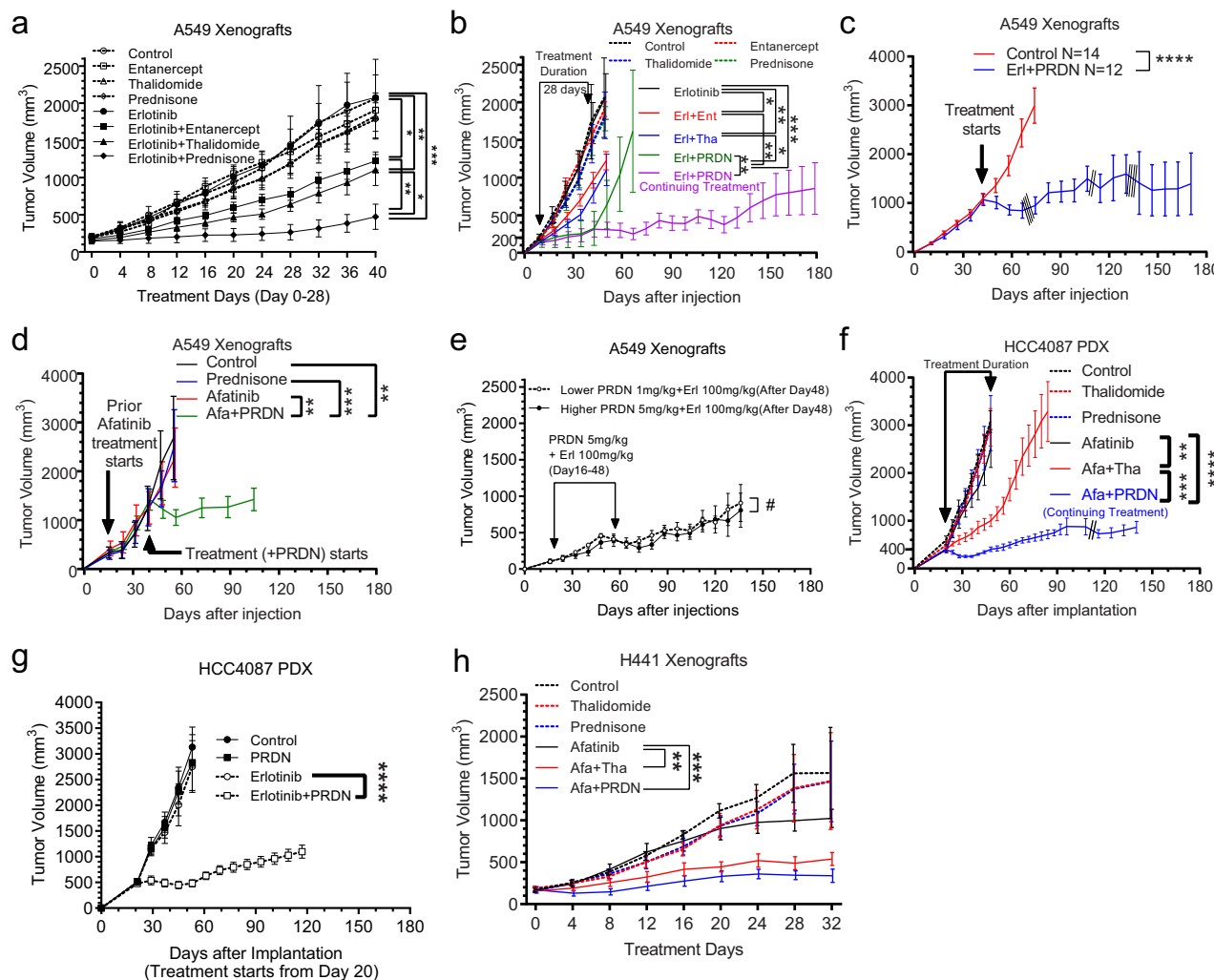

**Fig. 4 In vivo xenograft models show highly potent suppression of tumor growth with combined use of erlotonib and prednisone. a, b** Athymic mice were subcutaneously injected with 1 million A549 cells. After tumors became visible (200 mm³), mice were divided into eight groups (n = 8) and received the indicated treatment by oral gavage except Etanercept by Intraperitoneal injection. The dosages are: etanercept 3 mg/kg/day, erlotinib 100 mg/kg/day, thalidomide 150 mg/kg/day, Prednisone (PRDN) 5 mg/kg/day. The treatment lasted for 28 days, except for 4 mice in prednisone + erlotinib groups received drugs for around 6 months, while treatment was withdrawn from the other 4 mice in the same group. Tumor sizes were monitored every 4 days using a caliper. A and B are the same experiment with a different time range. **c** A549 xenograft tumors were allowed to grow to 1000 mm3 prior to starting treatment with control vehicle (n = 14), and a combination of prednisone 5 mg/kg/day plus erlotinib 100 mg/kg/day (n = 12). One mouse had to be sacrificed due to the tumor ulceration (slashed line). **d** A delayed start of prednisone. Twenty-four nude mice bearing A549 xenografts were divided into four groups (n = 6). Afatinib 25 mg/kg/day was given when tumors reaches 200 mm³. When tumors reach around 1500 mm³, prednisone 5 mg/kg/day was also given as indicated. **e** 12 mice bearing A549 xenografts received concurrent prednisone + erlotinib (doses same as above) from 200 mm3 to 500 mm³. Then, prednisone dose in 6 mice (n = 6) was reduced from 5 to 1 mg/kg/day, with erlotinib unchanged. **f** An EGFRwt PDX HCC4087, was implanted into NOD-SCID mice. After tumors reached 400 mm³, mice were divided into six groups and treated with indicated drugs using the same doses as above (n = 6). Treatments lasted for 28 days, except afatinib + prednisone (4 months). **g** A similar experiment (n = 8), using erlotinib. **h** A similar using H441 cells (n = 8). Tumor sizes are shown as mean volume ± SEM. *p < 0.05, **p < 0.01, ***p < 0.001, ****p < 0.0001, #p > 0.05 by repeated measures (RM) two-way ANOVA, adjusted by Bonferroni's test. The statistical analysis was performed on Graphpad Prism 9.0.0. Source data are provided as a Source Data file. p = 0.04,0.006,1E−4,0.003,0.03 (**a**); 0.04,0.006,1E−4,0.002,0.003,0.03 (**b**); 2E−6 (**c**); 0.003,5E−4,0.006 (**d**); 0.57 (**e**); 0.002.0.001,5E−5 (**f**); 5E−7 (**g**), 0.003,0.002 (**h**).

EGFR active mutant lung adenocarcinoma patients, who would be treated with EGFR TKIs (Fig. 7g). Moreover, either high cytokine or high RTK ligands signature also confers a worse prognosis in this group of patients (Fig. 7h, i).

## Discussion

Targeted therapy of cancer generally involves inhibition of a specific oncogenic pathway that plays a key role as an oncogenic driver in the pathogenesis or maintenance of cancer and has been most successful in the presence of activating mutations in RTK pathways that result in accentuated signaling[5,6]. Secondary resistance to targeted treatments includes the appearance of genetic mutations in the tumors usually detected after months or years of treatment. A number of studies have also reported that inhibition of receptor tyrosine kinase (RTK) pathways triggers rapid signaling changes in cancer cells. This cellular reprogramming has been reported in multiple cancer types and in response to the inhibition of multiple nodes in RTK signaling pathways. For example, inhibition of EGFR,

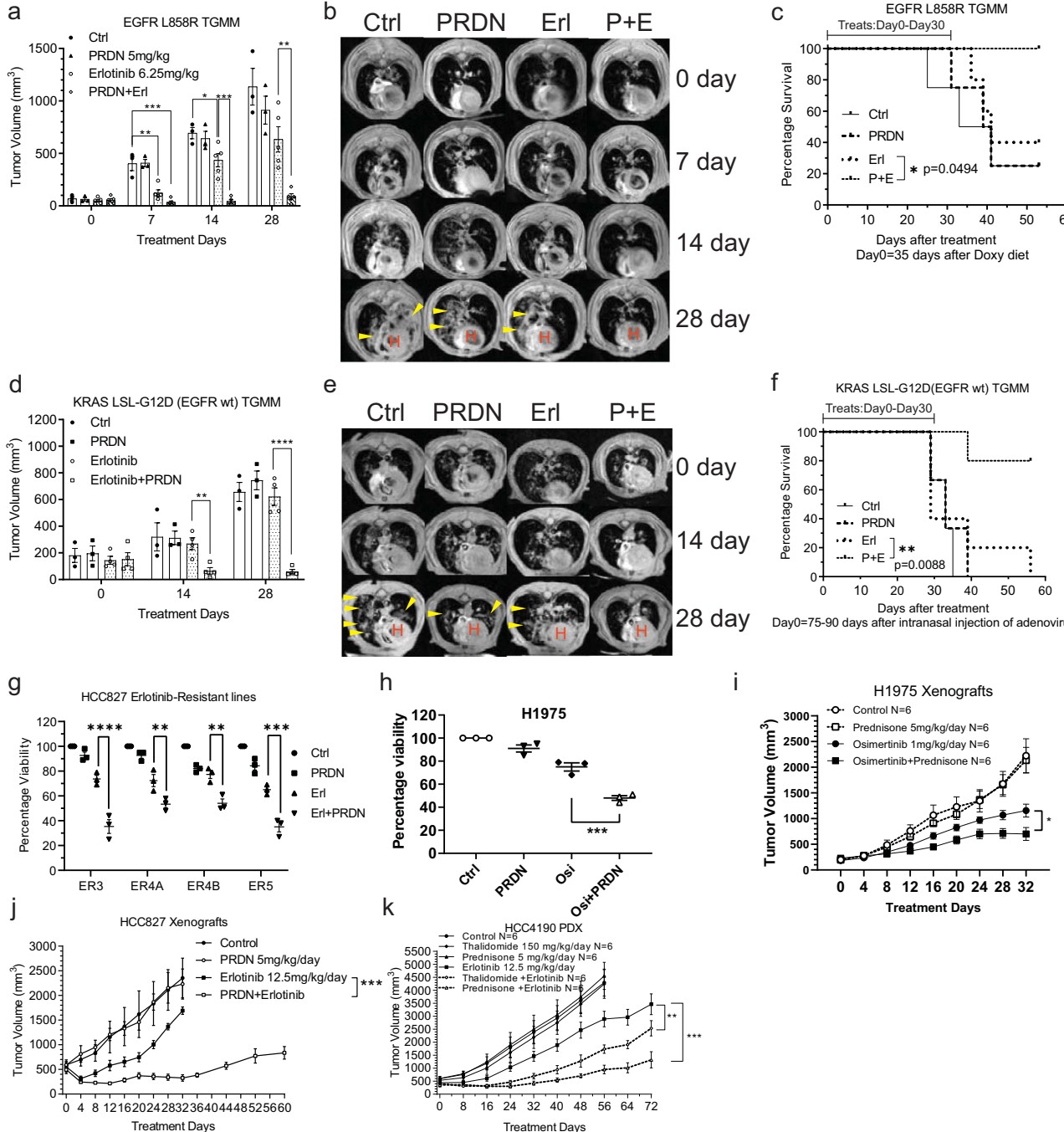

**Fig. 5 Synergistic effect of EGFR TKI plus prednisone in immunocompetent transgenic mouse models (TGMM) and secondary resistant EGFR mutant models. a–f** EGFR L858R mutant transgenic mice and KRAS LSL-G12D mutant (EGFRwt) transgenic mice were generated as described in Methods. They were divided into four groups each strain ($n = 3$ for **a** and $n = 3$ for **d** without erlotinib; $n = 4$ for **a** and $n = 5$ for **d** with erlotinib) and received 5 mg/kg/day prednisone and erlotinib (6.25 or 100 mg/kg/day for EGFR mutant or wt) for 28 days. Tumor growth was monitored at the indicated days by magnetic resonance imaging (MRI). Representative images were shown. The white round structure at the bottom are hearts, labeling with "H". Tumors are represented by white infiltrations around the hearts. Some representative tumors are pointed by arrows. Tumor sizes were calculated by ImageJ. The survival was recorded and drawn as Kaplan–Meier (KM) plots. **g** HCC827 derived erlotinib-resistant lines ER3, ER4A, ER4B, and ER5 were treated with 10 μM prednisolone and/or 1 μM erlotinib for 72 h. AlamarBlue assay was used to determine cell viabilities. $N = 3$ biological repeated experiments. **h** Similar experiment on EGFR L858R and T790M mutant H1975 cell line, where 10 nM osimertinib was used instead of erlotinib. $N = 3$ biological repeated experiments. **i, j** 1 million H1975 cells and 2 million HCC827 cells were s.c. injected into nude mice. Mice were divided into four groups ($n = 6$) and received the indicated treatments. **k** EGFR L858R mutant HCC4190 PDX were implanted into NOD-SCID mice. Mice were treated with the indicated drugs ($n = 6$ for each group). Xenograft and PDX sizes were measured by a caliper and shown as mean volume ± SEM. *$p < 0.05$, **$p < 0.01$, ***$p < 0.001$, ****$p < 0.0001$, by repeated measures (RM) two-way ANOVA, adjusted by Bonferroni's test (ADIJK), two-way ANOVA adjusted by Bonferroni's test (GH), and log-rank test (CF). The statistical analysis above was performed on Graphpad Prism 9.0.0. Source data are provided as a Source Data file. $p = 0.005,3e-6,0.03,5e-7,4e-5$ (**a**);$0.004,1e-6$ (**d**);$3e-5,0.006,0.003,1e-4$ (**g**),$2e-5$(**h**),$0.04$ (**i**),$0.0008$ (**j**),$0.08,0.004$ (**k**).

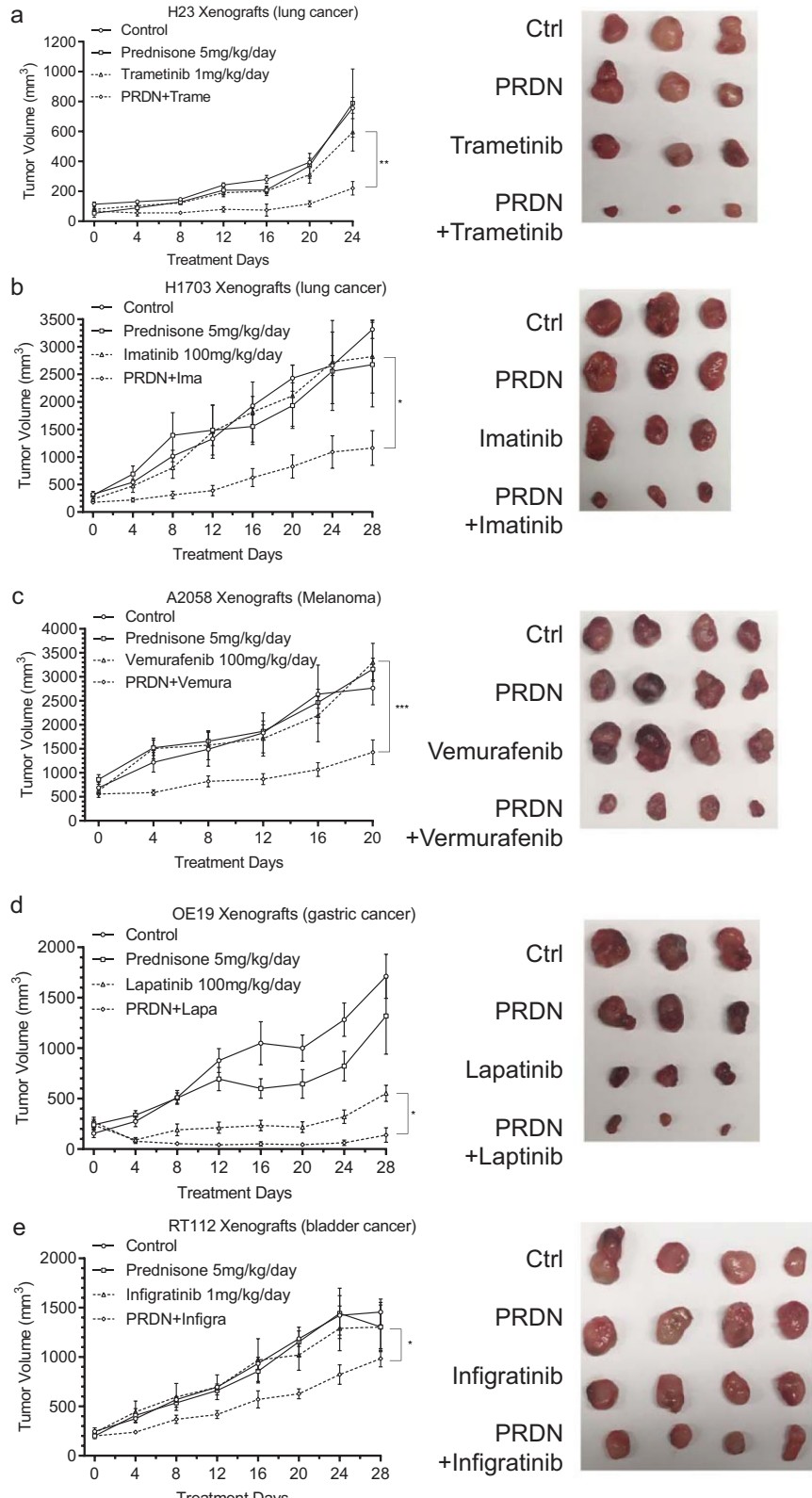

MEK or BRAF are reported to trigger compensatory signaling. This compensatory or adaptive response is, at least in large part homeostatic, and results in a resumption of previously interrupted signals or triggering of functionally similar signals. However, although this has not been carefully studied, the adaptive response may also result in the generation of ectopic signals not normally generated by the inhibited signaling pathway[4], and such signals may create a potential for synthetic lethality. There is substantial evidence from experimental models that the early adaptive response is biologically significant. However, it has been difficult to validate the importance of this adaptive response in patients since these signals are triggered within hours or days of treatment and it is difficult to obtain tissue at such early time points.

**Fig. 6 Tumor suppressive effects of a combination of other RTK pathway inhibitors and prednisone in xenograft studies. a–e** $N = 6$ mice per group. Ten million KRAS and ATM mutant H23 lung cancer cells, 5 million PDGFR amplification H1703 lung cancer cells, 1 million BRAF V600E mutant A2058 melanoma cells, 2 million ERBB2 amplification esophageal adenocarcinoma (gastric cancer) OE19 cells, and 5 million FGFR mutant RT112 bladder cancer cells were s.c. injected to 24 mice for each line. After tumor formation, mice were treated with the indicated drug for the indicated days. Trametinib is a MEK inhibitor and H23 is sensitive to it. Imatinib is a BCR-ABL/PDGFR inhibitor and H1703 is moderately sensitive/resistant. Vemurafenib targets BRAF V600E but A2058 is completely resistant. Lapatinib is an ERBB2 inhibitor and OE19 is sensitive. Infigratinib is an FGFR1-3 inhibitor under clinical trials and RT112 is sensitive. Lower dose (1 mg/kg/day) or higher dose (100 mg/kg/day) RTK inhibitors were used based on the initial sensitivities of these different cell lines. Representative tumor photos were shown. Tumor sizes were measured by a caliper and shown as mean volume ± SEM. *$p < 0.05$, **$p < 0.01$, ***$p < 0.001$, ****$p < 0.0001$, by repeated measures (RM) two-way ANOVA, adjusted by Bonferroni's test. The statistical analysis above was performed on Graphpad Prism 9.0.0. Source data are provided as a Source Data file. $p = 0.002, 0.05, 0.004, 0.03, 0.04$.

The adaptive response to inhibition of RTK signaling pathways is broad and multifaceted. For example, a study of BRAF inhibition in malignant melanoma found activation of at least 6 signaling pathways. Other studies have reported broad changes in gene expression and post translational modification of proteins among other changes. However, since oncogene addicted cells are very sensitive to pathway inhibition, adaptive responses are generally ineffective in protecting these cells from cell death. For example, EGFR inhibition is highly effective in most EGFR mutant NSCLC patients. However, secondary resistance develops inevitably in these initial responders[53]. Thus, in tumors that are initially sensitive to targeted inhibition, the biological significance of the early adaptive response may be that it leads to the survival of subsets of tumor cells that allow for the emergence of secondary resistance clones. Thus, the primary goal of studying adaptive responses to EGFR inhibition in EGFR mutant NSCLC has been the prevention of secondary resistance. However, although experimental studies have demonstrated that secondary resistance can be prevented by targeting of specific components of the adaptive response, whether such strategies will work in the clinical setting is unknown. Another interesting application of targeting aspects of the adaptive response could be to re-sensitize resistant cells to the RTK pathway inhibitor.

The adaptive response may also play an important role in mediating a primary resistance to RTK inhibition. For example, we have proposed that TNF and Type I IFN signaling play a key role in generating a primary resistance to EGFR inhibition in NSCLCs that express EGFRwt. It is known that EGFRwt expressing NSCLC generally exhibit a primary resistance and do not respond clinically to EGFR inhibition. Our recent data indicate that this primary resistance in EGFRwt NSCLC is mediated by a biologically significant adaptive response. Importantly, we have found that this adaptive response can be overcome therapeutically rendering NSCLCs that express EGFRwt sensitive to EGFR inhibition[26]. Since EGFRwt is expressed in the majority of NSCLC, a therapeutic targeting of this adaptive response would expand the use of EGFR inhibition to the majority of lung cancer. It is known that KRas mutant cancers are not dependent on EGFR signaling. EGFRwt is expressed in KRas mutant cancers and cell lines like A549 and H441. Although these KRas tumors and cell lines are resistant to EGFR inhibition, when the EGFRwt is inhibited in these KRas mutant tumors and in A549 and H441 cell lines, there is a robust compensatory/adaptive response as we have demonstrated in previous studies[26,27] and also in the present. Furthermore, the adaptive response in KRas mutant EGFRwt lines/tumors is biologically significant and a combined inhibition of EGFR+adaptive response renders these KRas mutant tumors sensitive to EGFR inhibition in experimental models as reported previously and shown in this study[26,27]. We propose that, EGFR inhibition may not work in these tumors because an adaptive survival mechanism triggered by EGFR inhibition negates its effect. A combined inhibition of EGFR+adaptive response either unmasks a requirement for

EGFR signaling for survival and/or sets up synthetic lethal conditions[27]. Also, as noted previous studies have shown that ERBB receptors play an important role in driving the growth of KRAS mutant tumors[51,52].

If the adaptive response is biologically significant what is the best approach to blunting it? The broad nature of the adaptive response to EGFR and other RTK targets, poses a formidable therapeutic challenge and raises the question whether it can be effectively and durably targeted using a single or a few drugs specifically designed to block one or several pathways. The alternative approach would be to use a relatively non-specific drug that would inhibit multiple components of the adaptive response. Our data indicate that the latter approach may be more effective.

Previous studies, including work from our laboratory, have indicated that it is possible to cripple the adaptive response in experimental models by targeting a single component of the adaptive response. It has also been demonstrated that a comprehensive targeting of the adaptive response, for example, by using the CDK7/11 inhibitor THZ1 which acts as a transcriptional repressor is more effective in preventing secondary resistance in experimental models[25]. Our studies examining the adaptive response to EGFR inhibition in lung cancer has focused on the role of TNF. We examined the efficacy of inhibiting TNF using specific inhibition vs. an inhibition of TNF plus additional inflammatory signals. For these experiments, we used etanercept, a specific TNF inhibitor. We also used thalidomide, a known inhibitor of TNF that also inhibits other inflammatory cytokines. Finally, we used prednisone, a known inhibitor of TNF that is also a broad inhibitor of inflammation. We found that prednisone is significantly more effective compared to etanercept in blocking the adaptive transcriptional response to EGFR inhibition in lung cancer cells, while thalidomide has an intermediate effect.

Prednisone may be a useful drug in combination with RTK pathway inhibition because of its broad suppression of RTK bypass signaling and inflammatory cytokines triggered by EGFR or other RTK target inhibition. We find similar results with dexamethasone suggesting that this is likely a general property of GCs. In particular, prednisone is remarkably effective in blunting the RTK bypass signaling that is triggered by inhibition of EGFR or other RTK signaling pathway nodes. The suppression of bypass RTK signaling was not detected with the other TNF inhibitor drugs tested in our study such as etanercept or thalidomide. It is important to note that RTK ligands and inflammatory cytokines correlate with a negative prognosis in EGFR mutant NSCLC. Prednisone is also a powerful suppressor of inflammatory cytokines and signaling. This suppressive effect is mediated largely by a broad suppression of transcription of both cytokines and RTK ligands (Supplementary Fig. 19). Previous studies have shown that EGFR mutations in NSCLC can activate STAT3 via an IL-6 dependent pathway[54–56]. Importantly, inhibition of EGFR also leads to activation of STAT3 via an IL-6 pathway[15]. As for YAP, an extensive cross talk between EGFR signaling and YAP has

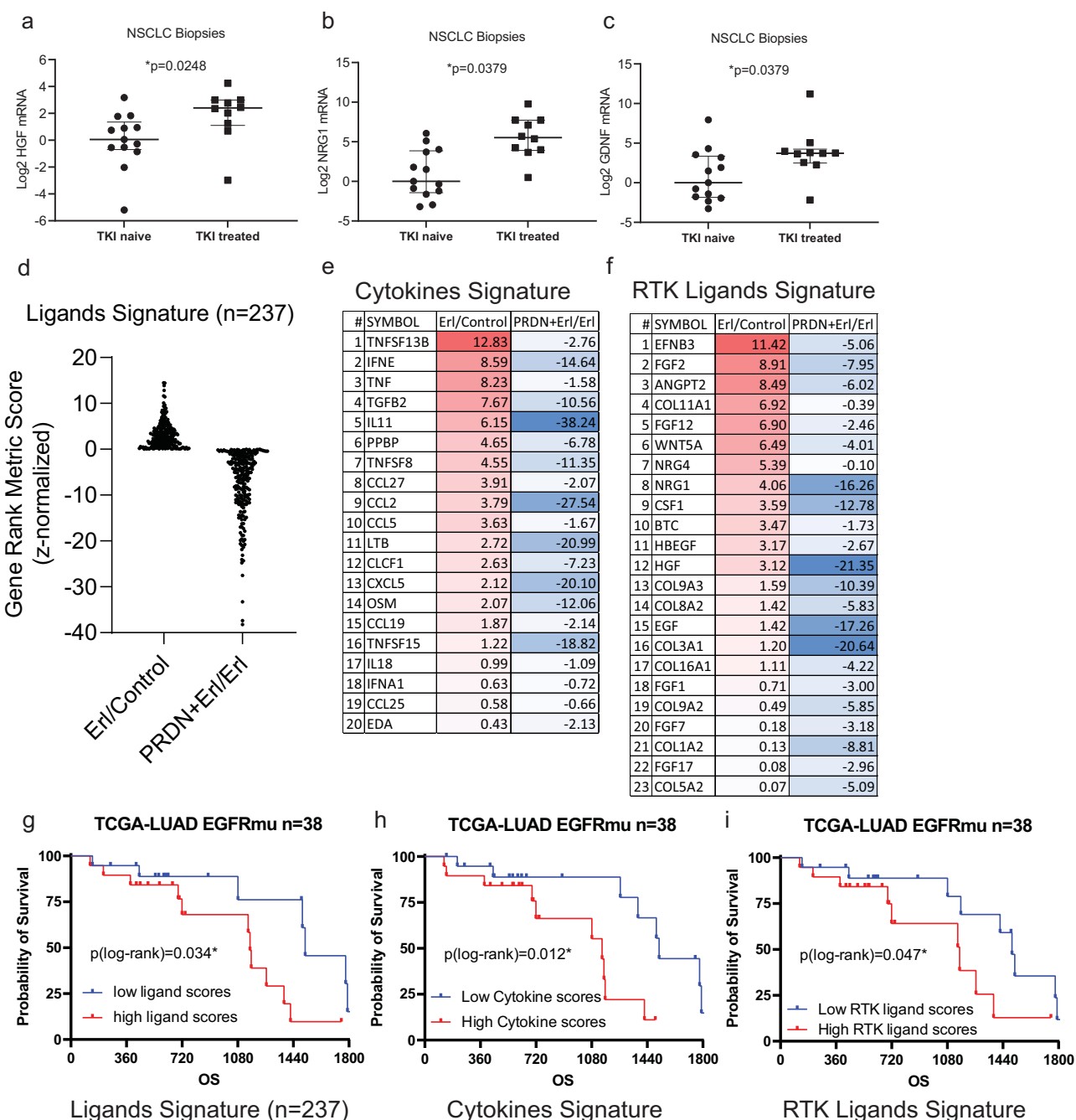

**Fig. 7 Assessment of RTK ligands in NSCLC tumor tissue. a–c** mRNA was extracted from FFPE tissue from 22 NSCLC patients ($n = 13$ untreated and $n = 9$ treated with erlotinib; 12 from UT Southwestern Medical Center, 10 from The Jackson Laboratory). HGF, NRG1, and GDNF mRNA levels were quantified by qPCR. Each dot represents the log2 mRNA level of each patient and median ± interquartile range. *$p < 0.05$, by Kolmogorov–Smirnov (KS) test. **d–f** From our A549 RNAseq result andj analyzed by GSEA, there are 237 ligand genes upregulated by erlotinib (positive z-score of Erl/Ctrl) and such upregulation was blocked by prednisolone (negative z-score of prednisolone + erlotinib/erlotinib). The human ligands list generating was described in the "Methods" section. Among these 237 ligands, there are 20 cytokines and 23 receptor tyrosine kinase ligands (RTKLs). These three sets of ligands formed three signatures. **g–i** RNAseq data of Fragments Per Kilobase of transcript per Million mapped reads (FPKM) among 38 patients with EGFR-activating mutant (exon 19 deletion or L858R) lung adenocarcinoma (LUAD) were downloaded from The Cancer Genome Atlas Program (TCGA)- Lung Adenocarcinoma (LUAD) database: portal.gdc.cancer.gov. 15004 genes out of the whole 60483 genes, with more than 19 (50%) non-zero expression among 38 cases, were selected to perform Single-sample Gene Set Enrichment Analysis (ssGSEA). The 3 signatures' scores for each of 38 patients were calculated. One score represents how the signature of ligands is coordinately up- or down-regulated in one case. For each signature, patients were equally divided by 50% low ($n = 19$) and 50% high scores ($n = 19$). Overall survival analysis were performed on two subgroups of patients, divided by 237 ligands, 20 cytokines, and 23 RTKLs. Kaplan–Meier (KM) plot was drawn, log-rank test was used to validate if these erlotinib-induced and prednisolone-blocked ligands would predict a worse prognosis in EGFR mutant NSCLC patients, who would receive EGFR inhibition treatment. *$p < 0.05$. by log-rank test. The Kolmogorov–Smirnov tests and survival analyses were performed by GraphPad Prism 9.0.0. Source data are provided as a Source Data file.

been reported[57]. The EGFR signaling pathway positively regulates YAP via MEK-ERK signaling and YAP signaling contributes to resistance to EGFR inhibition[58–60]. Prednisone also effectively suppresses key downstream signals such as STAT3 and YAP, implicated in mediating resistance to EGFR TKIs.

Consistent with the broad effects of prednisone on suppressing the adaptive response to RTK inhibition, experimental studies in mouse models show that the efficacy and durability of prednisone in combination with EGFR TKIs or other RTK inhibitors far exceeds other drugs such as etanercept and thalidomide. GCs may be used to treat side effects of cancer therapies, raising the question that if prednisone or other GCs are used to treat side effects of EGFR TKIs, a beneficial effect of combining EGFR inhibition plus prednisone would have quickly become apparent. However, it is unclear how commonly GCs are actually used to treat EGFR TKI induced side effects and when they are, it is usually used for short courses[61]. Our data clearly indicate that a sustained use of EGFR TKIs plus prednisone is required to suppress tumor growth. These data indicate that a combination of EGFR TKIs plus prednisone for extended periods has not really been tested in cancer patients. While prednisone has a number of side effects and can induce significant problems with sustained use, it continues to be widely used in multiple inflammatory and rheumatologic conditions because it is highly effective. We propose prednisone as a drug that can effectively inhibit the broad adaptive response to RTK inhibition with acceptable toxicity.

## Methods

**Cell lines**. A549 and A2058 were purchased from the American Type Culture Collection. OE19 was purchased from Millipore-Sigma. RT112 was purchased from DSMZ-German Collection of Microorganisms and Cell Cultures GmbH. HCC827/ER3, HCC827/ER4(A), and HCC827/ER5 were provided by Dr Trever Bivona (University of California, San Francisco). HCC827/ER4(B) was provided by Dr Eric Haura, Moffitt Cancer Center (Tampa, FL). All other NSCLC cell lines including H441, H2122, PC9, H3255, H23, and H1703 were from the Hamon Center for Therapeutic Oncology Research at UT Southwestern Medical Center. A2058 was cultured in DMEM containing 10% FBS. OE19 and RT112 were cultured in RPMI-1640 containing 10% FBS. All other cell lines were cultured in RPMI-1640 containing 5% FBS. Cell lines were authenticated by DNA fingerprints for cell-line individualization using Promega Stem Elite ID system, a short tandem repeat (STR)-based assay, at UT Southwestern Medical Center Genomics Core. Cells were tested for Mycoplasma contamination using an e-Myco kit (Boca Scientific).

**Drugs**. For the in vitro experiments, Erlotinib (10483), Afatinib (11492), Osimertinib (16237), Thalidomide (14610), Prednisolone (20866), Dexamethasone (11015) were obtained from Cayman Chemical. For the in vivo experiments, Prednisone (20677) and Thalidomide (14610) were purchased from Cayman Chemical. Erlotinib (E-4997), Afatinib (A-8644), Lapatinib (L-4899), Vemurafenib (V-2800), Trametinib (T-8123), Imatinib (I-5577), and Infigratinib (I-1500) were purchased from LC Laboratories. For both in vitro and in vivo usages, etanercept (Enbrel) (411175) was purchased from McKesson Medical Supply. Prednisolone is the active metabolite of prednisone. In vivo, prednisone is converted to prednisolone by the liver. Prednisolone was used for in vitro experiments and prednisone was used for in vivo experiments.

**Western blot and antibodies**. Western blot was performed according to standard protocols[62]. Western blot results were representative of at least three independent experiments. EGFR (06–847) antibody was from EMD Millipore; p-EGFR (Tyr1068) (2236), p-ERBB2 (Tyr1221/1222) (2243), ERBB2 (4290), p-ERBB3 (Tyr1289) (4791), ERBB3 (12708), p-RET (Tyr905) (3221), RET (14556), YAP (14074), p-STAT3 (Tyr705) (9145), STAT3 (9139), IκBa (4814), NIK (4994), RELB (10544), GR (47411), and p-GR (Ser211)(4161) antibodies were from Cell Signaling Technology; p-YAP (Tyr357) (ab62751) was from Abcam; β-actin (sc-47778) was from Santa Cruz Biotechnology.

**Real-time PCR and primers**. RNA was isolated by TRIzol Reagent (Thermo Fisher). cDNA reverse transcription was performed by High-Capacity cDNA Reverse Transcription kit (Applied Biosystems). Each PCR reaction was carried out in triplicate in a 20-μl volume using SYBR Green Master Mix (Applied Biosystems) for 15 min at 95 °C for initial denaturing, followed by 40 cycles of 95 °C for 15 s and 60 °C for 60 s in a ViiA 7 real-time PCR system (Applied Biosystems). Values for each gene were normalized to expression levels of ACTB (β-actin) mRNA. Actin primer pair was 5-CATGTACGTTGCTATCCAGGC-3 (forward), 5-CTCCTTAA

TGTCACGCACGAT-3 (reverse); TNF primer pair was 5-CCCAGGGACC TCTCTCTAATCA-3 (forward), 5-GCTACAGGCTTGTCACTCGG-3 (reverse); IκBa/NFKBIA primer pair was: 5-CGGCCTGGACTCCATGAAAG-3 (forward), 5-CCTTCACCTGGCGGATCACT-3 (reverse); VEGFD primer pair was: 5-ATC CCATCGGTCCACTAGGT-3 (forward), 5-TCCACGCACGTTTCTCTAGG-3 (reverse); These primers were synthesized by Millipo-Sigma. All other primers (RELB, NIK/MAP3K14, EGF, HBEGF, EREG, NRG1-4, HGF, GDNF, EFNA1, EFNA2, EFNA5, CSF1, FGF1-2, GAS6, PROS1, PDGFA, PDGFB, COL4A3, COL4A4, COL5A2, COL6A3, ANGPT1-2, NGF, BDNF, NTF3, VEGFC, WNT5A) used were KiCqStart SYBR Green Primers (KSPQ12012) predesigned by Millipo-Sigma. The first design for each target was used, excepted for GDNF using the third design. The sequences of these primers were provided by Millipo-Sigma on the datasheets.

**Cell viability assay**. AlamarBlue assay kit (DAL1025) was purchased from Thermo Fisher. Cells were cultured in Corning 96-well black plates with clear bottom and treated with drugs. Seventy-two hours later, AlamarBlue dye was added to each well according to the protocol and the plate was read by the POLARstar Omega Microplate Reader (BMG LABTECH) (excitation at 544 nm and emission at 590 nm). In the colony formation assay, cell lines were planted in six-well plates and treated with drugs for 14 days. Cell colonies were fixed by 100% methanol and then stained by 0.5% crystal violet in 25% methanol. Images captured by a scanner are representative of at least three independent experiments.

**Luciferase reporter assay**. Cells were plated in 24-well dishes followed by transfection with NF-κB or GRE7.3 luciferase plasmid and Renilla luciferase plasmid, by lipofectamine 2000. After 24 h, medium was changed, and drugs were added for another 24 h. A dual-luciferase reporter assay system (Promega) was used according to the manufacturer's protocol. Firefly luciferase activity was measured in the POLARstar Omega Microplate Reader (BMG LABTECH) and normalized based on Renilla luciferase activity. NF-κB luciferase plasmid was provided by Ezra Burstein (UT Southwestern). GRE7.3 luciferase plasmid was provided by Yamamoto lab (University of California, San Francisco).

**RNA-seq**. Total RNA was isolated by TRIzol Reagent (Thermo Fisher). RNA quantity check and RNA-seq was performed by 1. RNAseq was run with 150-bp paired-end reads (PE150) to produce ~40–50 million reads per sample. Sequencing data were further processed at UT Southwestern Medical Center Bioinformatics Core Facility. The raw reads were processed by Fastp[63] as the quality control tool, and STAR[64] as the alignment tool for mapping and generating gene counts. Differential expression and pathway analysis was performed based on EdgeR[65], GSEA[66] and ssGSEA[66].

By the Gene Set Enrichment Analysis (GSEA), all gene' ranking metric scores in the indicated comparisons were calculated, followed by z-score normalization using the median and median absolute deviation (MAD) method. The difference between each gene score and median score of all genes was calculated by subtraction. MAD was yielded as the median of all absolute differences. As MAD does not equal to 0 in each comparison, the z-score of each gene was calculated by subtracting the median score from the gene score and then divided by 1.486*MAD.

The P value was calculated by the formula: (1-NORMSDIST(ABS("z-score"))) *2 in Excel. The critical z-score at 95% confidence interval (CI) are ±1.96. The p-value associated with a 95% CI is 0.05.

The false discovery rate (FDR) was controlled at 0.05 by the Benjamini-Hochberg (BH) procedure. P values for all genes (number = n) were re-ranked from gene with the largest p (rank = n) to smallest p (rank = 1). For each gene (rank = i), FDR(i)= P(i)*n/i. Then, by stepwise calculation from gene (i-1) to gene (1), FDR(i) = minimal value of FDR(i) and FDR(i + 1).

The human ligands list used for generating signatures was based on the compendium of ligand-receptor pairs[67], by combining ligands from three recent databases of SingleCellSignalR[68], CellTalkDB[69], and NATMI[70].

**ELISA**. HGF ELISA kit was purchased from R&D (DHG00B); NRG1-beta1 ELISA kit was purchased from Thermo Fisher (EHNRG1). Cells were cultured in serum-free medium and treated with indicated drugs for 72 h. Cell lysates were collected by RIPA buffer. Protein concentrations were determined by Pierce BCA Protein Assay kit (Thermo Fisher). ELISA was performed according to the manufacturer's protocol.

**Phospho-RTK array**. Proteome Profiler Human Phospho-RTK Array (R&D, ARY001B) was performed according to the protocol of this kit. Firstly, cancer cells cultured in 10 cm dishes were treated as indicated in the figures and legends. Cell lysates were collected by the lysis buff provided in the kit and quantified as in Western Blot protocol above. Raw images were captured by ChemiDoc MP Imaging System (Bio-Rad) at varying exposure times to detect the highest and lowest expressing ranges of RTKs. Raw images (.scn) were then processed by Image Lab 6.0.1 (Bio-Rad) to generate final images (.tif) shown in figures. Final images (.tif) were automatically analyzed by (Western Vision Software) to quantitate each dot's pixel density representing this phospho-RTK expressing level. In the figures shown in this study, some representative p-RTKs were framed up and footnoted.

The quantitation was based on the longest exposure, except for the framed p-RTKs in shorter exposures to avoid overexposure on these high expressing p-RTKs. RTKs were ranked by basal expression. The log 2-fold changes of every treatment conditions comparing to control were show in the heatmap.

**Transcription factor array.** Forty-five transcription factors' activities were examined by Cignal 45-Pathway Reporter Array (Qiagen). Cells were planted in the 96-well plates provided in this kit, following the transfection, treatment of indicated drugs and luciferase measurement according to the protocol of this kit.

**Cell line-Derived Xenograft experiments.** Female nude mice (strain: 088) at 4–6-weeks old were purchased from Charles River Laboratories. Cancer cells were injected subcutaneously into the flanks of nude mice. After tumors form and reach the indicated size at treatment Day 0, mice were randomly divided into different groups, receiving indicated drugs as described in figures and legends. Tumor volumes were calculated by the formula: volume $= 0.5 \times$ length $\times$ width $\times$ width. Mice were killed when tumors reached >20 mm in length, after the indicated last treatment, or serious shrinkage-caused ulceration happened. Number of mice in each group was shown in figures and legends.

**PDX experiments.** The EGFRwt and KRAS G13C mutant HCC4087 and EGFR L858R mutant HCC4190 tissues were originally surgically obtained as Passage 0 (P0) from 2 NSCLC patients at UT Southwestern, with Institutional Review Board (IRB) approval and informed consents. These mutations were identified by exome sequencing. Female NOD/SCID mice (strain: 394, Charles River Laboratories) at 4–6-weeks old were purchased to bearing PDXs. In the PDX implantation surgery on mice, PDX tumor tissues were cut into small pieces ($\sim 20\,mm^3$) and subcutaneously implanted into NOD-SCID mice of serial generations (P1, P2 and so on). P4 tumor-bearing NOD-SCID mice were used in this study.

**Transgenic mouse models.** EGFR mutant transgenic mouse model (TGMM) (TetO-EGFR-L858R and CCSP-rtTA) were gifted from Katerina Politi at Yale University (New Haven, Connecticut, USA) and transported to UT Southwestern for this study. Mice were given continuous doxycycline in diet (TD.01306 2018 625 Doxycycline, Envigo) to induce tumors. Lung tumor formation was confirmed by comparison of MRI between 4 and 5 weeks after doxycycline induction. EGFRwt and KRAS mutant TGMM mice (LSL-Kras G12D) (008179) were purchased from Jackson Laboratories. The colony was expanded by breeding heterozygous LSL-Kras G12D mice with wild-type mice. Genotyping was performed per the protocol on Jackson Laboratories' website. The KRas-mutant lung cancer was driven by intranasal injected $2.5 \times 10^8$ plaque-forming units of Adeno-CMV-Cre(University of Iowa)-mediated induction of KRas G12D expression[50]. Treatments were initiated once the tumors were confirmed by with MRI at about 10–12 weeks after tumor induction.

All animals in this study were housed at 65–75 °F, 40–60% humidity, and 12 light /12 dark cycle.

**MRI scanning.** MRI was conducted to scan tumors in TGMM, at UT Southwestern Medical Center Mouse MRI Core, Advanced Imaging Research Center, using a 7T small animal MRI scanner (Bruker) equipped with a 40-mm quadrature radio-frequency coil (ExtendMR). Under anesthesia by inhalation of 1.5–3% isoflurane mixed in with medical-grade oxygen via nose cone, the animals were placed supine on a mouse holder, with a pneumatic respiratory sensor and electrocardiography electrodes for cardiac sensing, headfirst with the lung centered with respect to the center of the radiofrequency coil. Mice chests were shaved and conducting hydrogels were applied to optimize contact between electrodes and mouse. All MRI acquisitions were gated using both cardiac and respiratory triggering. The bore temperature was kept at 23 ± 2 °C to assure adequate and constant heart rate. Two-dimensional scout images on three orthogonal planes (transverse, coronal and sagittal) were acquired to determine positioning. Lower-resolution gradient echo (T1_FLASH) images were acquired on transverse plane to fine-adjust slice position. Finally, higher-resolution gradient-echo images were recorded on the transverse plane, with major parameters as follows: repetition time was 200 ms (actual was changing according to electrocardiography R-R interval, in range 200–240 ms), echo time was 1.966 ms, flip angle was 45°, number of average was 12, field of view was $32 \times 32\,mm^2$, matrix size was $256 \times 256$, slice number was 17–21 (changed upon mouse lung size) and slice thickness was 1 mm without any gap. Image analyses were performed using ImageJ bundled with Java 1.8.0_112. MRI images were processed and analyzed by two blinded researchers using ImageJ (NIH).

**Patient data.** Patient formalin-fixed, paraffin-embedded (FFPE) tissues were obtained from the Jackson Laboratory ($n = 10$) or UT Southwestern Medical Center ($n = 13$). Ten patients were treated with TKI prior to the tissue collection, while 13 were never treated before. The RNA from these FFPE tissues was extracted by RecoverAll Total Nucleic Acid Isolation Kit (AM1975, Thermo Fisher), and followed by real-time PCR described above. TCGA patient data were downloaded from portal.gdc.cancer.gov. 38 TCGA-LUAD patients (any stages) with overall survival (OS) data, and harboring classical TKI-sensitive mutations, L858R or exon

19 deletion, but without T790M mutation, were selected in this study. The following bioinformatic analysis was described in the figure legends.

**Study approval.** All animal studies were performed under Institutional Animal Care and Use Committee-approved protocols at UT Southwestern Medical Center and North Texas VA Medical Center. Patient tissues and medical records were obtained from UT Southwestern Medical Center and Jackson Laboratory with IRB approval.

**Statistics.** Each dot represent one result from three independent experiments. If one experiments was performed, one dot or one value would be shown without statistical analysis. Error bars represent the mean of three experiments ± SEM of unless indicated otherwise. Combination effects were analyzed by two-way ANOVA with Bonferroni's correction to adjust the significance level for multiple comparisons. For time course, one-way ANOVA with Dunnett's test was used. When comparing multiple factors' differences between two samples, two-sided $t$-test was corrected by two-stage step-up method of Benjamini, Krieger and Yekutieli, with the desired FDR was set as 0.05. Kaplan–Meier survival curves were constructed and compared by log-rank test. Experiments on patient samples were shown as median ± interquartile range, analyzed by Kolmogorov–Smirnov test. Statistical methods for RNA-seq data only were described in RNAseq method part. All tests were two-sided.

**Reporting summary.** Further information on research design is available in the Nature Research Reporting Summary linked to this article.

## Data availability

RNA-seq data that support the findings of this study was deposited in the Sequence Read Archive PRJNA763241. Source data are provided with this paper.

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

## Acknowledgements

This work was supported in part by funding from the Department of Veteran's Affairs to AH (2I01BX002559-07) and from the National Institutes of Health (1R01CA244212-01A1) to A.H. This work was also supported by NCI Lung Cancer SPORE (P50CA70907), U01CA176284, and CPRIT (RP110708 and RP160652) to J.D.M. D.E.G. is supported by a National Cancer Institute (NCI) Midcareer Investigator Award in Patient-Oriented Research, K24CA201543-01. S.B. is supported by grants from the National Institutes of Health (RO1CA197796) and the National Aeronautics and Space Administration (NNX16AD78G). E.A.A. is supported by CPRIT RR160080, NCI 2P50CA070907-22, Welch Foundation grant 1975-20190330, A Breath of Hope Lung Foundation Fellowship Award (ABOHLF 2020), NCCN Foundation Young Investigator Award (NCCN 2021), and Forbeck Foundation Grant. D.Z. was supported by NIH grant R01 CA194578, Research reported in this publication was supported in part by the Harold C. Simmons Comprehensive Cancer Center's Biomarker Research Core, which are supported by NCI Cancer Center Support Grant 1P30 CA142543-03. We acknowledge NIH shared instrumentation grant 1S10OD023552-01 that funded the MRI equipment.

## Author contributions

K.G., G.G. and A.A.H. designed experiments. K.G., G.G., N.A.B., X.Y. and Y.Z. E.A.A. performed or assisted with experiments. D.E.G., J.D.M. provided cell lines, PDX or human tissue specimens. K.G., G.G., D.E.G., S.B., D.Z. E.A.A. and A.A.H. analyzed data. K.G. and A.A.H. wrote the manuscript. Study conception and supervision was conducted by A.A.H.

## Competing interests

The Department of Veteran's Affairs has filed a patent for the combined use of EGFR inhibitors plus prednisone in lung cancer with Amyn Habib as the inventor. The other authors declare no competing interests.
