## [Peer Review File · Nature Communications]

Comprehensive targeting of resistance to inhibition of RTK signaling pathways by using glucocorticoidsReviewers' comments:

Reviewer #1 (Remarks to the Author):

In this manuscript, the authors focused on EGFRwt and EGFRmut non-small cell lung cancer (NSCLC) and investigated that efficacy of a broad and focused inhibition of adaptive response triggered by EGFR-tyrosine kinase inhibitor (TKI) treatment. In this study, they used three TNF inhibitors, etanercept, thalidomide and prednisone/prednisolone. Etanercept is a TNF signaling-specific inhibitor and thalidomide and prednisone/prednisolone inhibit multiple inflammatory responses. They found that prednisone/prednisolone, a glucocorticoid, most effectively inhibited production of many inflammatory cytokines including TNF, many ligands of receptor tyrosine kinase (RTK) and bypass activation of various RTKs induced by EGFR-TKI in both EGFRwt and EGFRmut NSCLC cells. As the results, prednisone/prednisolone rendered these NSCLC cells sensitive to EGFR-TKI. Additionally, they report prednisone/prednisolone can suppress growth of various type tumors, as well as NSCLC. These findings are interesting to overcome clinical problems which are widely observed in EGFR-TKI monotherapy. But critical point of this manuscript is lacking the mechanisms why prednisone/prednisolone combined with EGFR inhibitor works better.

Major comments

Introduction:

Information of the three drugs, etanercept (Enbrel), thalidomide and prednisone, that were described in the Results section should be briefly introduced in the Introduction section.

Methods:

The authors should provide more detailed information of RNA-seq data analysis. For example, methods for quality control of reads, reads mapping and estimation of gene expression. Additionally, the authors should cite references of computer programs used in this study. For example, edgeR, GSEA, ssGSEA.

In addition, accession number to access RNA-seq data produced in this study should be provided. If the authors have not obtained accession number, please deposit the RNA-seq sequence data in a public nucleotide sequence database, such as Sequence Read Archive (SRA), The European Nucleotide Archive (ENA), DDBJ Sequence Read Archive (DRA) or related databases.

Results:

The authors showed the percentage of a number of blocked erlotinib-induced genes (Fig. 1B). According to the figure legend, the threshold to select upregulated genes was set at 1.05-fold changes. I think the threshold was extremely low. Were these genes significantly upregulated by erlotinib treatment? Please provide p-value and FDR corresponding the threshold.

What is the meanings of denominator and numerator in the "Erl-induced" column in Figure 1D? For example, 111/112up in Inflammatory Response and 89/90up in Receptor tyrosine kinase (RTK). The meanings of the denominator and of the numerator should be provided in the figure legend.

The band intensity between control and Erl is not so significantly different in Figure 2H and 2I, but in Fig 2J-2M, the band intensity between control and Erl is so different. Why the intensity of bands of control lanes and Erl were so different even in the same cell lines?

The authors described that prednisolone and prednisone were used in vitro and in vivo experiments respectively in the Method section. However, the authors described as "To inhibit the non-canonical activation of NF- κ B, prednisone downregulates NF- κ B subunits such as RELB, and ... (Fig. 2P-U)" in Page 6, line 20-23, and described "blocked by PRDN" in corresponding the figure legend. The authors described PRDN is abbreviation of prednisolone in legend of Figure 1. Which is correct, prednisone or prednisolone? In addition to this paragraph, there are many confusing usages of prednisone and prednisolone. Please check the words throughout the text.

In supplementary table 1 and 2, the authors should describe what GSEA score mean. Is it fold-

change or NES?

Please check the q-value obtained by Bonferroni test in Figure 3 and Supplementary Figure 2. In general, Bonferroni-corrected p-value is not smaller than original p-value.

The authors should provide accurate scale bar in Figure 4 and Supplementary Figure 3. Positive values are not fairly evaluated in the figures. Scale bar must include all data range and each data cells should be colored more properly. (-3 to -11 was all same in white)

The authors said "The RTKs upregulated by erlotinib and suppressed by prednisone include MET, ERBB2, ERBB3, and RET among others" in Page 8, line 11-12. According to Figure 4, however, signals of RTK arrays and log₂FC values of the heatmap of both cell lines are not greater than those of control. In A549, for example, log₂FC values of Erl-3d MET, Erl-3d ERBB2, Erl-3d ERBB3 and Erl-3d RET are -1.2, -4.4, -1.5 and -0.1, respectively. Similar data are also found in HCC827. I do not think these RTKs are not upregulated by erlotinib-treatment. It might not be adequate to say that these RTKs are upregulated by erlotinib-treatment.

The authors described "prednisone completely suppressed the adaptive activation of STAT3 and YAP (Fig. 5C-G)" in Page 8, line 25-26. However, some cell lines lack STAT3 data. I suggest that phosphorylation of STAT3 should be examined by western blot for all cell lines.

Although the authors showed the combination of erlotinib plus prednisone had the strongest effect on tumor growth suppression and remains more than 6 months tumor suppression effect (Fig. 6), tumor growth curve obtained from long-time course administration of erlotinib plus prednisone is only shown. Have the authors investigated long-time tumor suppression effects of combination of erlotinib plus etanercept and of erlotinib plus thalidomide in vivo?

The authors claim that "the combination erlotinib plus prednisone to be highly effective in inhibiting the growth of this PDX tumor for an extended period of time (Figure 6F)". However, corresponding figure shows the results of afatinib treatment. Which is correct, erlotinib or afatinib? If afatinib is correct, please mention a reason why the authors use afatinib not erlotinib for PDX model.

The authors claim that "we tested H1975 cells (with EGFR L858R/T790M) using afatinib and found that these cells also could be rendered sensitive to EGFR inhibition if prednisolone is used (Fig. 7H)". T790M mutation is a resistant mutation prevent 1st and 2nd generation EGFR-TKI, erlotinib and afatinib, from binding to EGFR. However, the result indicates that H1975 cells survive against afatinib by activating bypass RTK signaling. Is it true?

The authors also claim that "we examined whether a combination of erlotinib plus prednisone would be effective in treating H1975 tumors in vivo. H1975 cells were injected We found that a combined treatment with afatinib and prednisone resulted in a highly effective suppression of tumor growth (Fig. 7I)". Which is correct, erlotinib or afatinib? If afatinib is correct, have the authors investigated whether the same effect is similar in vitro and in vivo experiment using erlotinib?

Discussion:

The authors indicated that various RTKs, STAT3 and YAP are activated by EGFR-TKI and the activations are effectively suppressed by prednisone treatment in many NSCLC cell lines. However, the relationships between these RTKs and STAT3/YAP in the cells used in this study are not clear. The authors need to discuss what kind of biological mechanisms activate STAT3 and YAP in the NSCLC cells.

Minor comments

The sub-numberings of Figure 3 are wrong. Please correct the sub-numberings of the figure.

Drug names described in the text (Page 9, line 21) and in legend of Supplementary Figure 8 are not consistent. Please make these consistent.

In Supplementary Figure 8I-K, the authors should provide more clear figures.

In Figure 9, The authors should cite reference of ligand-receptor pair database. The reference is mentioned in <https://github.com/LewisLabUCSD/Ligand-Receptor-Pairs>.

Reviewer #2 (Remarks to the Author):

The authors have shown that suppression of TNF signals (particularly by prednisolone) suppresses cancer cell adaptation and enhances the efficacy of molecular-targeted drugs. The experiments are well planned and carried out, and the manuscript is well written. However, several issues listed below should be addressed.

Major comments

1. Authors demonstrated that various RTKs are activated and adapted in various types of tumors, including EGFR mutant lung cancer, KRAS mutant cancer, HER2-amplified gastric cancer, BRAF mutant cancer, and PDGFRA mutant lung cancer, to adapt targeted drugs, such as EGFR inhibitor, MEK inhibitor, HER2 inhibitor, and PDGFR inhibitor. The results in this study suggested that this phenomenon may be caused by activation of the NF-KB pathway when the driver signal is blocked. On the other hand, it is well accepted that in KRAS mutated A549 cells, even if EGFR is blocked, mutated KRAS maintains the downstream signals, so that the EGFR inhibitor does not work. In addition, in EGFR-T790M-positive H1975 cells, it is explained that the EGFR inhibitor does not work because the EGFR signal is not suppressed because the EGFR inhibitor (erlotinib) does not inhibit the mutant EGFR. Therefore, it is necessary to explain the mechanism why erlotinib causes an adaptation reaction in KRAS mutant cancer and EGFR-T790M-positive lung cancer.

2. Prednisolone was used at 5 mg/kg and 1 mg/kg in this study. Generally, long-term use at 20 mg/body/day or more causes various side effects such as immunosuppression, impaired glucose tolerance, and osteoporosis. It is desirable to explain how much doses of prednisolone used in this study corresponds to in humans. Also, can prednisolone of 1 mg/kg or less enhance the effect of molecular-targeted drugs?

Reviewer #3 (Remarks to the Author):

Gong et al. show that combined inhibition of EGFR (Erlotinib) and broad acting TNF inhibitors (Prednisone) in models of NSCLC could improve therapeutic efficacy compared to EGFR inhibition alone, or in combination with more specific TNF inhibition. While patients with NSCLC may initially respond to EGFR/RTK inhibitors relapse is common and so the authors are addressing an important clinical problem. The authors previously identified TNF upregulation in response to EGFR inhibition. Now, the authors show that various pathways related to the adaptive signalling response from RTK inhibitors were downregulated (shown by mRNA, protein, transcription factor activity) in cancer cells treated with both RTK inhibitors and TNF inhibition. The in vitro data is backed up nicely by in vivo experiments and patient data. The animal experiments are very thorough, conducted both in immunocompetent and xenograft models. These experiments were conducted long term in lung cancer models and in multiple other tumour types and show impressive tumour suppression. The inclusion of data from patients treated with TKIs adds strength to their conclusions. The authors provide accurate description of methods and use appropriate statistics. The authors should be commended on the volume of work that has been completed.

However, in my opinion this volume of data sometimes detracts from the main points of the paper, and the presentation of the results could be improved. This would enhance the ability to interpret

the data and for readers to understand the main points and significance of the work.

1. Figure 1 and 2 could be condensed and merged together. I think perhaps Figure 1B could be merged with the data in Figure 2. Figure 1C is slightly confusing because TNF signalling, regulation of TNFR1 signaling, TNFR1-induced NFkappaB signaling is unchanged with Entanercept or Thalidomide co-treatment alongside Erlotinib, yet it's shown nicely that TNF mRNA/protein and NFkB is downregulated by these drugs in Figure 2B-G.
2. For Figure 3 I think the tables showing the means, p and q values are an unnecessary representation of the data considering the graph shows the data clearly. The authors could consider putting this table in the supplementary figures. Perhaps the authors could also highlight important genes with an arrow?
3. Figure 4 and 5 could be condensed and merged together. This would be much better represented in one figure as the heatmap from the RTK array plus the key result backed up by western blot (perhaps with only the important timepoints and/or quantitation to reduce volume of data and make the conclusion clearer). The rest can be made into supplementary material.
4. The colour scale for the heatmaps in Figure 4 is poorly distributed, the red covers 0 to -2 and leaves -2 to -11 all as white. I also find it difficult to determine the signalling trends in the heatmaps by eye throughout the paper when the fold changes are displayed as numbers and colours I would recommend to just display colours as standard for heatmaps.
5. Figure 6E, was there a no treatment or Erl only control?
6. Figure 7B, could the authors maybe draw arrows to the tumours or put a dotted line around the heart to distinguish the heart and tumour a bit easier?
7. Figure 7, is there data past 50-60 days post treatment start now? If so, it would be good to update in revisions.
8. For Figure 9G-I, could the authors state how many patients were in the low and high groups of each comparison?

Minor:

1. Figure 8, it would be helpful to include the type of tumour next to the cell line name to determine each tumour type without looking at the figure legend or text.
2. I encourage the authors to reread the manuscript carefully for some small writing errors (eg. page 12, second line of last paragraph "could be rendered sensitive by to EGFR inhibition by..." should be "could be rendered sensitive to EGFR inhibition by...").
3. While the authors are considering the cell survival/drug resistance promoting effects of TNF signalling in this paper, it would be interesting to see the cancer cell lines sensitivity to the cytostatic/cytotoxic effects of TNF. I suspect these cancer cell lines are insensitive to the cytostatic/cytotoxic, and that sensitising to TNF in combination with TKIs could also promote tumour suppression, while activating anti-tumour immune responses (eg. Michie et al. CIR 2018, DOI: 10.1158/2326-6066.CIR-18-0428). This could theoretically lead to a better long-term response to TKI compared to TKI suppressing inflammation. I know combinations of BRAF/MEK inhibition with checkpoint blockade has been investigated in melanoma, however I'm not sure the extent of research for TKIs in NSCLC combined with immune checkpoint blockade. I don't expect the authors to add this data as it is outside the scope of this manuscript, but could be worth discussing and at the very least this comment may give the authors an additional research direction to consider.

Overall the manuscript is well written and contains a lot of work. If the data was more succinct and easily interpretable I believe this paper would be impactful for NSCLC and across multiple cancer types.

Response to Reviewer Comments

Reviewer #1 (Remarks to the Author):

In this manuscript, the authors focused on EGFRwt and EGFRmut non-small cell lung cancer (NSCLC) and investigated that efficacy of a broad and focused inhibition of adaptive response triggered by EGFR-tyrosine kinase inhibitor (TKI) treatment. In this study, they used three TNF inhibitors, etanercept, thalidomide and prednisone/prednisolone. Etanercept is a TNF signaling-specific inhibitor and thalidomide and prednisone/prednisolone inhibit multiple inflammatory responses. They found that prednisone/prednisolone, a glucocorticoid, most effectively inhibited production of many inflammatory cytokines including TNF, many ligands of receptor tyrosine kinase (RTK) and bypass activation of various RTKs induced by EGFR-TKI in both EGFRwt and EGFRmut NSCLC cells. As the results, prednisone/prednisolone rendered these NSCLC cells sensitive to EGFR-TKI. Additionally, they report prednisone/prednisolone can suppress growth of various type tumors, as well as NSCLC. These findings are interesting to overcome clinical problems which are widely observed in EGFR-TKI monotherapy. Critical point of this manuscript is lacking the mechanisms why prednisone/prednisolone combined with EGFR inhibitor works better.

Response:

We thank Reviewer 1 for a thoughtful and detailed critique of our manuscript. Addressing the points made by Reviewer 1 has allowed us to improve the manuscript and to add clarity to the paper. We appreciate the comment about our findings being relevant to overcome widely observed problems in EGFR TKI monotherapy. Please note that there is some reorganization of figures in this revised manuscript based on recommendations made by Reviewer 3.

Why prednisone/prednisolone combined with EGFR inhibitor works better.

Summary of response: The compensatory/adaptive response to EGFR inhibition promotes therapeutic resistance and has two major facets.

1. Activation of inflammatory signals. This component is blocked by all 3 drugs (etanercept/enbrel, thalidomide, and prednisone as shown in Fig. 1).
2. Activation of bypass RTK signaling. This component is only blocked by prednisone and not by etanercept or thalidomide (Supplementary Table 2, Supplementary Fig. 1A, and Supplementary Fig. 4).

Many studies have shown that bypass RTK signaling is a key resistance mechanism and we propose that **because prednisone is more effective because it blocks both arms of the adaptive response** while the other drugs block only one arm of the adaptive response as shown in the Schematic below (included in the manuscript as Supplementary Fig. 18)

Legend to Supplementary Fig. 18: EGFR inhibition results in activation of inflammatory pathways and bypass RTK signaling. Both arms of the adaptive response promote therapeutic resistance. In control vehicle treated cells, both arms of the adaptive response are active and the tumor is resistant to EGFR TKIs. The inflammatory arm of the adaptive response is blocked by etanercept, thalidomide, or prednisone. Thus, etanercept and thalidomide treatment results in a partial response to EGFR inhibition. However, only prednisone blocks the bypass RTK activation and thus is a more effective suppressor of the adaptive response to EGFR inhibition and results in a more effective suppression of tumor growth.

Detailed response:

This is an important point. A central component of the adaptive compensatory response to EGFR inhibition is bypass RTK signaling. Thus, if EGFR is inhibited other receptor tyrosine kinases take over the function of the EGFR. When the EGFR or a downstream signaling component is inhibited, cancer cells fight back with increased secretion of ligands for multiple receptors^{1, 2}. This leads to the activation of multiple receptors that take over the function of the EGFR^{3, 4, 5, 6}. This is bypass RTK signaling, an adaptive change widely accepted as a central mechanism of resistance to EGFR inhibition^{7, 8}. The suppression of bypass RTK inhibition and its mechanisms are extensively described in the paper (Fig. 2A, D, Supplementary Fig. 2A-B, Supplementary Table 1B-C, Supplementary Table 2, Fig. 2B-C, E-F, Supplementary Fig. 2C-F, Fig. 3, Supplementary Fig. 3, Supplementary Fig. 4). The other drugs etanercept and thalidomide have no effect on RTK bypass signaling and only suppress inflammatory signals. Thus, we propose that combination of prednisone plus EGFR inhibition works better because prednisone induces a widespread shutdown of bypass RTK signaling while etanercept and thalidomide do not (Supplementary Fig. 1A, Supplementary Fig. 4, Supplementary Table 2).

Please note that the previous Supplementary Fig. 3 is now Supplementary Fig. 4 In the revised manuscript.

The mechanism is as follows. Prednisone induces a shutdown of bypass RTK signaling, because it is a broad transcriptional repressor^{9, 10, 11} (etanercept and thalidomide are not). Prednisone induces a broad and almost universal transcriptional repression of RTK ligands resulting in suppression of RTK inhibition (Fig. 2-3, Supplementary Table 1B-C). RTK ligands (growth factors) play a key role in mediating resistance to EGFR inhibition. Prednisone blocks this surge of growth factors that result from EGFR inhibition while etancercept (enbrel) and thalidomide do not (Supplementary Fig. 1A, Supplementary Fig. 4, Supplementary Table 2).

Major comments

Comment 1/Introduction:

Information of the three drugs, etanercept (Enbrel), thalidomide and prednisone, that were described in the Results section should be briefly introduced in the Introduction section.

Response: We have added information about the three drugs to the introduction as recommended.

Comment 2/Methods:

The authors should provide more detailed **information of RNA-seq data analysis**. For example, methods for quality control of reads, reads mapping and estimation of gene expression. Additionally, the authors should cite references of computer programs used in this study. For example, edgeR, GSEA, ssGSEA.

Response: We have added detailed RNAseq information in the Methods section. We used Fastp as the quality control tool, and STAR as the alignment tool for mapping and estimation of gene expression. The references of computer programs were also cited. Detailed information is provided in the Methods section.

Comment 3/Methods: In addition, accession number to access RNA-seq data produced in this study should be provided. If the authors have not obtained **accession number**, please deposit the RNA-seq sequence data in a public nucleotide sequence database, such as Sequence Read Archive (SRA), The European Nucleotide Archive (ENA), DDBJ Sequence Read Archive (DRA) or related databases.

Response: We will deposit the RNA seq data upon acceptance of the manuscript or in response to a request from the reviewer. We have deposited RNA seq data for our recent Nature Cancer study¹².

Comment 4/Results:

The authors showed the percentage of a number of blocked erlotinib-induced genes (Fig. 1B). According to the figure legend, **the threshold to select upregulated genes** was set at (oring0.07=1.05FC)(1=2FC, 2=4FC,p=0.05)1.05-fold changes. I think the threshold was

extremely low. Were these genes significantly upregulated by erlotinib treatment? Please provide p-value and FDR corresponding the threshold.

Response: We have now used the z-normalized GSEA ranking metric score and set the threshold as 1.96, which corresponds to $p = 0.05$, and is equivalent to a 2-fold change. Finally, we select significantly upregulated genes with z-score cutoff over 1.96 and FDR in Benjamini-Hochberg method <0.05 . The detailed information about calculation were provided in the Methods section. All genes' original GSEA scores, z-scores, p-values, and FDR can be provided upon request.

Although a fewer number of genes is altered as the stricter cutoffs were used, the results are similar and the trend is the same (Fig. 1B). Prednisone results in a greater suppression of the EGFR inhibition driven transcriptional changes compared to etanercept or thalidomide.

Comment 5:

What is the meanings of **denominator and numerator** in the “Erl-induced” column in Figure 1D? For example, 111/112up in Inflammatory Response and 89/90up in Receptor tyrosine kinase (RTK). The meanings of the denominator and of the numerator should be provided in the figure legend.

Response: Fig. 1D is now Supplementary Fig. 1A. We have provided the denominator and numerator in the figure legend. 111/122 up means that of the 122 inflammatory pathways analyzed by GSEA, 111 are upregulated by erlotinib.

Comment 6:

The **band intensity** between control and Erl is not so significantly different in Figure 2H and 2I, but in Fig 2J-2M, the band intensity between control and Erl is so different. Why the intensity of bands of control lanes and Erl were so different even in the same cell lines?

Response: We provide a quantitative analysis of relative band intensity by doing **densitometry readings** to clarify the band intensities in for Figure 2H-I and J-M to help clarify this issue. When we examine the quantification normalized to actin, the reduction in IKB α induced by erlotinib is signal is similar. Please note that the data are now shown in Fig. 1J-O.

We would point out that with varying exposure times in different experiments, and depending on whether the primary antibody is being reused, there may be minor differences from experiment to experiment.

Comment 7: The authors described that **prednisolone and prednisone** were used in vitro and in vivo experiments respectively in the Method section. However, the authors described as “To inhibit the non-canonical activation of NF-kB, prednisone downregulates NF-kB subunits such as RELB, and ... (Fig. 2P-U)” in Page 6, line 20-23, and described “blocked by PRDN” in corresponding the figure legend. The authors described PRDN is abbreviation of prednisolone in legend of Figure 1. Which is correct, prednisone or prednisolone? In addition to this

paragraph, there are many confusing usages of prednisone and prednisolone. Please check the words throughout the text.

Response: We apologize for the confusion and have clarified this in the text and each figure legend. Prednisolone is the active metabolite of prednisone. Prednisolone is the drug used for all in vitro studies, prednisone is used for all in vivo studies (the liver converts prednisone to prednisolone).

Comment 8: In supplementary table 1 and 2, the authors should describe **what GSEA score mean.** Is it fold-change or NES?

Response: GSEA score means Genes Ranking Metric Score, given by GSEA software. It is not fold-changes, nor NES. It refers to the gene expression difference in the indicated comparison.

Comment 9: Please check **the q-value obtained by Bonferroni test** in Figure 3 and Supplementary Figure 2. In general, Bonferroni-corrected p-value is not smaller than original p-value.

Response: Indeed, in figure 3 (data now shown as Supplementary Table 1B-C), our Bonferroni-corrected p-values/q-values is bigger than the original p-values.

However, in Supplementary Figure 2, there was a typographical error. The p should be q, and q should be p, q is larger than p. We have corrected the error (data now shown in Supplementary Fig. 2A-B). The data values are shown in the Figure and we can also provide the original data if requested.

Comment 10: The authors should provide **accurate scale bar** in Figure 4 and Supplementary Figure 3. Positive values are not fairly evaluated in the figures. Scale bar must include all data range and each data cells should be colored more properly. (-3 to -11 was all same in white)

Response: The scale bar in Fig. 4 and Supplementary Fig. 3 and all figures with heatmaps have been revised to include all data range and each data cell is now colored properly according to the range. Our conclusions are unchanged.

Comment 11: The authors said “The **RTKs upregulated** by erlotinib and suppressed by prednisone include MET, ERBB2, ERBB3, and RET among others” in Page 8, line 11-12. According to Figure 4, however, signals of RTK arrays and log₂FC values of the heatmap of both cell lines are not greater than those of control. In A549, for example, log₂FC values of Erl-3d MET, Erl-3d ERBB2, Erl-3d ERBB3 and Erl-3d RET are -1.2, -4.4, -1.5 and -0.1, respectively. Similar data are also found in HCC827. I do not think these RTKs are not upregulated by erlotinib-treatment. It might not be adequate to say that these RTKs are upregulated by erlotinib-treatment.

Response: The key comparison is one day erlotinib treatment to 3 days erlotinib treatment. The typical pattern of bypass RTK activation is as follows. When EGFR is inhibited, activation of Met or ERBB2 is downregulated at **one day**, suggesting that the EGFR is driving activation of Met or ERBB2 (Fig. 3A, B, C, D). **At 72h** as part of the adaptive response bypass RTK

signaling occurs as Met or ERBB2 **become re-activated** because EGFR inhibition leads to an upregulation of ligands for Met and ERBB2. Importantly, the reactivation of Met and ERBB2 occurs while the EGFR is still inhibited as shown in the figure 3A-D. Thus, as the Reviewer correctly observes there is no difference between 0h and 72h, but there is a big difference between 24h and 72h (1 day and 3 days) as is typical for an adaptive response¹³. We quite agree that erlotinib treatment is not directly activating other RTKs. However, it is the well-known and extensively reported adaptive response to erlotinib mediated EGFR inhibition that triggers activation of other receptors such as Met or ErbB^{2, 7, 14, 15, 16}.

There is also another pattern of adaptive activation as noted for YAP (Fig. 3C-I). Here, there is no initial decrease in YAP activation by erlotinib, suggesting that the EGFR is not driving the activation of YAP under basal conditions, but there is an increase in YAP activation as part of the adaptive response and this activation is suppressed by prednisolone.

Comment 12: The authors described “prednisone completely suppressed the adaptive activation of STAT3 and YAP (Fig. 5C-G)” in Page 8, line 25-26. However, some cell lines lack **STAT3 data**. I suggest that phosphorylation of STAT3 should be examined by western blot for all cell lines.

Response: The STAT3 panels for all cell lines have been added to Fig. 3D-I. STAT3 is not activated in response to EGFR inhibition in HCC827 cells¹⁷. STAT3 is activated by EGFR inhibition in PC9 and H3255 cells but not in HCC827, A549 and H441 cell lines. We would point out that STAT3 is only one readout of the many signals triggered by EGFR inhibition.

Comment 13: Although the authors showed the combination of erlotinib plus prednisone had the strongest effect on tumor growth suppression and remains more than 6 months tumor suppression effect (Fig. 6), tumor growth curve obtained from long-time course administration of erlotinib plus prednisone is only shown. **Have the authors investigated long-time tumor suppression effects of combination of erlotinib plus etanercept and of erlotinib plus thalidomide in vivo?**

Figure 4A-B shows a comparison of the effect of etanercept (enbrel), thalidomide, or prednisone alone or in combination with erlotinib. The combination of etanercept plus erlotinib or thalidomide plus erlotinib is effective in suppressing tumor growth only for 30-60 days (Fig. 4B) after which tumors grow rapidly and mice have to be sacrificed per IACUC guidelines. Ethically, it is not possible to continue the experiment for up to six months. Prednisone on the other hand, in combination with erlotinib induces a powerful and sustained suppression of tumor growth for the duration of the experiment (6 months).

Comment 14: The authors claim that “the combination erlotinib plus prednisone to be highly effective in inhibiting the growth of this PDX tumor for an extended period of time (Figure 6F)”. However, corresponding figure shows the results of afatinib treatment. Which is correct, **erlotinib or afatinib?** If afatinib is correct, please mention a reason why the authors use afatinib not erlotinib for PDX model.

The data shown in Figure 6F are indeed derived from an afatinib experiment (now shown as Figure 4F). The text was incorrect in referring to erlotinib rather than afatinib. Both erlotinib, a first generation EGFR TKI (Fig. 4A-C, E) and afatinib, a second generation EGFR TKI (Fig. 4D, F, H) inhibit EGFR kinase activity and in combination with prednisone are highly effective in suppressing tumor growth. We have previously tested both afatinib and erlotinib in the specific HCC4087 PDX model to see if they would provide effective suppression, but in the original manuscript shown data only for afatinib since the results were quite similar. We have now added a panel to Fig. 4 showing the same effect of erlotinib in the same PDX model (Fig. 4G)

Comment 15 The authors claim that “we tested H1975 cells (with EGFR L858R/T790M) using afatinib and found that these cells also could be rendered sensitive to EGFR inhibition if prednisolone is used (Fig. 7H)”. **T790M mutation is a resistant mutation prevent 1st and 2nd generation EGFR-TKI, erlotinib and afatinib**, from binding to EGFR. However, the result indicates that H1975 cells survive against afatinib by activating bypass RTK signaling. Is it true?

Response: While the first generation TKI erlotinib is indeed ineffective against the T790M mutation, multiple studies have shown that the second generation EGFR TKI afatinib is effective in inhibiting the EGFR T790M mutation found in H1975 cells^{18, 19, 20, 21, 22, 23}. Our data also indicate that afatinib blocks the T790M mutated EGFR in H1975 cells and triggers an adaptive response.

Comment 16 The authors also claim that “we examined whether a combination of erlotinib plus prednisone would be effective in treating H1975 tumors in vivo. H1975 cells were injected We found that a combined treatment with afatinib and prednisone resulted in a highly effective suppression of tumor growth (Fig. 7I)”. Which is correct, **erlotinib or afatinib**? If afatinib is correct, have the authors investigated whether the same effect is similar in vitro and in vivo experiment using erlotinib?

Response: Thank you for pointing this out. Afatinib is correct, erlotinib is a typographical error in the results section. The figure (Fig. 7H-I, now Figure 5H-I) is correctly labeled. We have not done the experiment with erlotinib because it is known that erlotinib is ineffective in suppressing the kinase activity of EGFR harboring the T790M mutation^{24, 25}.

Discussion/ Comment 17 The authors indicated that various RTKs, STAT3 and YAP are activated by EGFR-TKI and the activations are effectively suppressed by prednisone treatment in many NSCLC cell lines. However, the relationships between these RTKs and STAT3/YAP in the cells used in this study are not clear. The **authors need to discuss** what kind of biological mechanisms activate STAT3 and YAP in the NSCLC cells.

Response: We are happy to add the following information to the discussion. The biological mechanisms that activated STAT3 and YAP have been well described in previous studies. Previous studies have shown that EGFR mutations in NSCLC can activate STAT3 via an IL-6 dependent pathway^{26, 27, 28}. Importantly, inhibition of EGFR also leads to activation of STAT3 via an IL-6 pathway¹⁷. As for YAP, an extensive cross talk between EGFR signaling and YAP has been reported²⁹. The EGFR signaling pathway positively regulates YAP via MEK-ERK

signaling and YAP signaling contributes to resistance to EGFR inhibition^{30, 31, 32}.

Minor comments

Comment: The sub-numberings of Figure 3 are wrong. Please correct the sub-numberings of the figure.

Response: Noted, done

Comment: Drug names described in the text (Page 9, line 21) and in legend of Supplementary Figure 8 are not consistent. Please make these consistent.

Response: We have corrected.

Comment In Supplementary Figure 8I-K, the authors should provide more clear figures.

We have provided more clear figures for the experiments in Supplementary Fig. 8I-K (now Supplementary Fig. 9I-K).

In Figure 9, The authors should cite reference of ligand-receptor pair database. The reference is mentioned in <https://github.com/LewisLabUCSD/Ligand-Receptor-Pairs>.

We have added the references of the databases in the Method section.

Reviewer #2 (Remarks to the Author):

General Comment: The authors have shown that suppression of TNF signals (particularly by prednisolone) suppresses cancer cell adaptation and enhances the efficacy of molecular-targeted drugs. The experiments are well planned and carried out, and the manuscript is well written. However, several issues listed below should be addressed.

Response:

We appreciate Reviewer comments that experiments are well planned and carried out and that the manuscript is well written.

Major comments

Comment 1. Authors demonstrated that various RTKs are activated and adapted in various types of tumors, including EGFR mutant lung cancer, KRAS mutant cancer, HER2-amplified gastric cancer, BRAF mutant cancer, and PDGFRA mutant lung cancer, to adapt targeted drugs, such as EGFR inhibitor, MEK inhibitor, HER2 inhibitor, and PDGFR inhibitor. The results in this study suggested that this phenomenon may be caused by activation of the NF-KB pathway when the driver signal is blocked. On the other hand, it is well accepted that in KRAS mutated A549 cells, even if EGFR is blocked, mutated KRAS maintains the downstream signals, so that the EGFR inhibitor does not work. In addition, in EGFR-T790M-positive H1975 cells, it is explained that the EGFR inhibitor does not work because the EGFR signal is not suppressed

because the EGFR inhibitor (erlotinib) does not inhibit the mutant EGFR. Therefore, it is necessary to explain the mechanism why erlotinib causes an adaptation reaction in KRAS mutant cancer and EGFR-T790M-positive lung cancer.

Response: KRAS mutant cancer: We acknowledge that KRas mutant cancers are not dependent on EGFR signaling. EGFR wild type is expressed in K-Ras mutant cancers and cell lines like A549 and H441. Although these KRas tumors and cell lines are resistant to EGFR inhibition, when the EGFR wild type is inhibited in these KRas mutant tumors and in A549 and H441 cell lines, there is a robust compensatory/adaptive response as we have demonstrated in previous studies^{12, 33} and also throughout the present study. Furthermore, the adaptive response in KRas mutant EGFR wild type lines/tumors is biologically significant and a combined inhibition of EGFR+adaptive response renders these KRas mutant tumors sensitive to EGFR inhibition in experimental models^{12, 33}. This is also extensively investigated in the current manuscript. We propose that the primary resistance to EGFR inhibition in EGFRwt and KRas mutant NSCLC does not necessarily indicate that EGFR signaling is irrelevant to the malignant phenotype. Rather, EGFR inhibition may not work in because an adaptive survival mechanism triggered by EGFR inhibition negates its effect. A combined inhibition of EGFR+adaptive response either unmasks a requirement for EGFR signaling for survival and/or sets up synthetic lethal conditions¹².

Importantly, two recent studies in Science Translational Medicine demonstrated that EGFR family receptors facilitate KRas driven tumorigenesis^{34, 35}. Both studies used A549 cells as a major experimental model^{34, 35}. The data in the current manuscript indicate that prednisone suppresses the adaptive response in KRas mutant cell lines (A549, H441) as well as multiple animal models and results in suppression of tumor growth when combined with EGFR inhibition. The signaling networks triggered by this EGFR inhibition induced adaptive response are the same in KRas mutant and KRas wt cells, i.e bypass RTK signaling and inflammatory responses being major components of this response.

EGFR T790M mutation: This mutation renders EGFR resistant to binding erlotinib and is a common mechanism of secondary resistance. Thus erlotinib is unable to inhibit EGFR tyrosine kinase activity. However, a number of studies have shown that the second generation EGFR TKI afatinib is effective in inhibiting the EGFR T790M mutation found in H1975 cells^{18, 19, 20, 21, 22, 23}. We used Afatinib not erlotinib to treat T790M (there was a typographical error in the results section that has been corrected). When the EGFR harboring the T790M mutation is inhibited, it responds in a similar way by triggering an adaptive response that is blocked by prednisone in vitro (Fig. 5H-I, Supplementary Fig. 12). The typographical error in the results section that mentioned erlotinib has been corrected to afatinib. Figure 5I is correctly labeled and show that the combination of afatinib plus prednisone is effective in H1975 cells harboring a T790M mutation in an animal model.

Comment 2. Prednisolone was used at 5 mg/kg and 1 mg/kg in this study. Generally, long-term use at 20 mg/body/day or more causes various side effects such as immunosuppression, impaired glucose tolerance, and osteoporosis. It is desirable to explain how much doses of

prednisolone used in this study corresponds to in humans. Also, can prednisolone of 1 mg/kg or less enhance the effect of molecular-targeted drugs?

Response: We thank Reviewer 2 for raising this important issue and we are happy to clarify. Certainly the side effects of prednisone are well known as it is a widely used drug for many inflammatory conditions. The dose of prednisone recommended for COVID-19 is 40mg/day. <https://www.covid19treatmentguidelines.nih.gov/immunomodulators/corticosteroids/>.

For diseases such as temporal arteritis a dose of up to 80mg/day are used³⁶. Typically, prednisone is started at a high dose and then tapered to a lower dose used for maintenance therapy. As pointed out by Reviewer 2, the safety limit for long-term use is about 20mg/day.

The dose used in our study is quite consistent with use in human patients for various diseases. We use 5mg/kg and 1mg/kg in our mouse models that are equal to 0.4mg/kg and 0.08mg/kg in human, considering an adult weight of 60 kg, the dose would be 24mg/day and 5mg/day.

Importantly, the prednisone dose can be tapered in our experimental paradigm to 1mg/day without a loss of efficacy (Fig. 4E), just as it is done in human patients. The dose calculation of mouse to human was performed as has been previously described³⁷.

Reviewer #3 (Remarks to the Author):

General Comments: Gong et al. show that combined inhibition of EGFR (Erlotinib) and broad acting TNF inhibitors (Prednisone) in models of NSCLC could improve therapeutic efficacy compared to EGFR inhibition alone, or in combination with more specific TNF inhibition. While patients with NSCLC may initially respond to EGFR/RTK inhibitors relapse is common and so the authors are addressing an important clinical problem. The authors previously identified TNF upregulation in response to EGFR inhibition. Now, the authors show that various pathways related to the adaptive signalling response from RTK inhibitors were downregulated (shown by mRNA, protein, transcription factor activity) in cancer cells treated with both RTK inhibitors and TNF inhibition. The in vitro data is backed up nicely by in vivo experiments and patient data. The animal experiments are very thorough, conducted both in immunocompetent and xenograft models. These experiments were conducted long term in lung cancer models and in multiple other tumour types and show impressive tumour suppression. The inclusion of data from patients treated with TKIs adds strength to their conclusions. The authors provide accurate description of methods and use appropriate statistics. The authors should be commended on the volume of work that has been completed.

However, in my opinion this volume of data sometimes detracts from the main points of the paper, and the presentation of the results could be improved. This would enhance the ability to interpret the data and for readers to understand the main points and significance of the work.

Response:

We really appreciate the multiple positive comments by Reviewer 3 and have tried to follow the recommendations to condense some of the data and make the work more accessible to the reader.

Comment 1. Figure 1 and 2 could be condensed and merged together. I think perhaps Figure 1B could be merged with the data in Figure 2. Figure 1C is slightly confusing because TNF signalling, regulation of TNFR1 signaling, TNFR1-induced NFkappaB signaling is unchanged with Entanercept or Thalidomide co-treatment alongside Erlotinib, yet it's shown nicely that TNF mRNA/protein and NFkB is downregulated by these drugs in Figure 2B-G.

Response: As per Reviewer 3 recommendations Figure 1 and 2 are combined and Figure 1C is removed. Figure 1D is now moved to Supplementary Fig. 1.

Comment 2. For Figure 3 I think the tables showing the means, p and q values are an unnecessary representation of the data considering the graph shows the data clearly. The authors could consider **putting this table in the supplementary figures.** Perhaps the authors could also highlight important genes with an arrow?

Response: We have put the tables showing the means, p and q values in Supplementary Table 1 as Supplementary Table 1 B-C. We have added arrows to highlight important genes in Figure 2A, 2D.

Comment 3. Figure 4 and 5 could be condensed and merged together. This would be much better represented in one figure as the heatmap from the RTK array plus the key result backed up by western blot (perhaps with only the important timepoints and/or quantitation to reduce volume of data and make the conclusion clearer). The rest can be made into supplementary material.

Response: Figure 4 and 5 were merged and reorganized as recommended (new Figure 3). Some data were moved to a new Supplementary Fig. 3.

Comment 4. The colour scale for the heatmaps in Figure 4 is poorly distributed, the red covers 0 to -2 and leaves -2 to -11 all as white. I also find it difficult to determine the signalling trends in the heatmaps by eye throughout the paper when the fold changes are displayed as numbers and colours I would **recommend to just display colours as standard for heatmaps.**

Response: We have revised the heatmaps to display colors as standard for all figures with heatmaps.

Comment 5. Figure 6E, was there a **no treatment or Erl only control?**

Response: This experiment was specifically designed to see if we could taper the dose of prednisone without a loss of efficacy. We are only comparing a high and a low dose of prednisone for the experiment in this figure. A comparison with no treatment and EGFR inhibitor only control is shown in Fig. 4A, 4B, 4D, 4F, 4G). Please note that the previous Figure 6 is now Figure 4.

Comment 6. Figure 7B, could the authors maybe draw arrows to the tumours or put a dotted line around the heart to distinguish the heart and tumour a bit easier?

Response: Done. Arrows were drawn to indicated tumors. “H” labels were put on hearts. Please note that Fig. 7 is now Fig. 5.

Comment 7. Figure 7, is there **data past 50-60 days** post treatment start now? If so, it would be good to update in revisions.

Response: Unfortunately, we do not have the data for longer timepoints. Starting from around 30 days, some mice became sick and died. And the experiment was terminated around 60 days because only the combination treatment group survived.

Comment 8. For Figure 9G-I, could the authors state **how many patients** were in the low and high groups of each comparison?

Response: 19 Vs 19. This is now mentioned in the figure legend. Please note that Figure 9 is now Figure 7.

Minor Comments:

Comment 1. Figure 8, it would be helpful to include the type of tumour next to the cell line name to determine each tumour type without looking at the figure legend or text.

Response: Done, Fig. 8 is now Fig. 6.

Comment 2. I encourage the authors to reread the manuscript carefully for some small writing errors (eg. page 12, second line of last paragraph “could be rendered sensitive by to EGFR inhibition by...” should be “could be rendered sensitive to EGFR inhibition by...”).

Response: we have corrected the error and also re-read the manuscript carefully for additional typographical errors.

Comment 3. While the authors are considering the cell survival/drug resistance promoting effects of TNF signalling in this paper, it would be interesting to see the cancer cell lines sensitivity to the cytostatic/cytotoxic effects of TNF. I suspect these cancer cell lines are insensitive to the cytostatic/cytotoxic, and that sensitising to TNF in combination with TKIs could also promote tumour suppression, while activating anti-tumour immune responses (eg. Michie et al. CIR 2018, DOI: 10.1158/2326-6066.CIR-18-0428). This could theoretically lead to a better long-term response to TKI compared to TKI suppressing inflammation. I know combinations of BRAF/MEK inhibition with checkpoint blockade has been investigated in melanoma, however I’m not sure the extent of research for TKIs in NSCLC combined with immune checkpoint blockade. I don’t expect the authors to add this data as it is outside the scope of this manuscript, but could be worth discussing and at the very least this comment may give the authors an additional research direction to consider.

Response: Thank you for the comment. This is certainly a fascinating and complex area. We have previously investigated the role of TNF in mediating resistance to EGFR inhibition, and found that inhibition of TNF in lung cancer cells synergized with EGFR inhibition in experimental lung cancer models³³. This does not necessarily conflict with the interesting study by Michie et al., since the effects of TNF in the immune system could be quite different. Whether TNF

targeting could synergize with checkpoint inhibitors seems unresolved. There is not much data available for lung cancer. Although experimental studies have suggested that TNF inhibition plus checkpoint inhibition may be effective in some experimental models^{38, 39}, patients who receive concomitant checkpoint inhibitors and TNF inhibitors may fare worse⁴⁰.

We thank the reviewer for these suggestions and have added some information to the discussion.

Comment: Overall the manuscript is well written and contains a lot of work. If the data was more succinct and easily interpretable I believe this paper would be impactful for NSCLC and across multiple cancer types.

Response: We are pleased that the reviewer values the work as potentially impactful. We have followed Reviewer 3 guidelines for making the paper more succinct and to add clarity.

References:

1. Harbinski F, *et al.* Rescue screens with secreted proteins reveal compensatory potential of receptor tyrosine kinases in driving cancer growth. *Cancer Discov* **2**, 948-959 (2012).
2. Wilson TR, *et al.* Widespread potential for growth-factor-driven resistance to anticancer kinase inhibitors. *Nature* **487**, 505-509 (2012).
3. Zhang Z, *et al.* Activation of the AXL kinase causes resistance to EGFR-targeted therapy in lung cancer. *Nat Genet* **44**, 852-860 (2012).
4. Engelman JA, *et al.* MET amplification leads to gefitinib resistance in lung cancer by activating ERBB3 signaling. *Science* **316**, 1039-1043 (2007).
5. Chandarlapaty S, *et al.* AKT inhibition relieves feedback suppression of receptor tyrosine kinase expression and activity. *Cancer Cell* **19**, 58-71 (2011).
6. Duncan JS, *et al.* Dynamic reprogramming of the kinome in response to targeted MEK inhibition in triple-negative breast cancer. *Cell* **149**, 307-321 (2012).
7. Sun C, Bernards R. Feedback and redundancy in receptor tyrosine kinase signaling: relevance to cancer therapies. *Trends in biochemical sciences* **39**, 465-474 (2014).
8. Rotow J, Bivona TG. Understanding and targeting resistance mechanisms in NSCLC. *Nat Rev Cancer* **17**, 637-658 (2017).

9. Barnes PJ. Anti-inflammatory actions of glucocorticoids: molecular mechanisms. *Clin Sci (Lond)* **94**, 557-572 (1998).
10. Webster JC, Cidlowski JA. Mechanisms of Glucocorticoid-receptor-mediated Repression of Gene Expression. *Trends Endocrinol Metab* **10**, 396-402 (1999).
11. Coutinho AE, Chapman KE. The anti-inflammatory and immunosuppressive effects of glucocorticoids, recent developments and mechanistic insights. *Mol Cell Endocrinol* **335**, 2-13 (2011).
12. Gong K, *et al.* EGFR inhibition triggers an adaptive response by co-opting antiviral signaling pathways in lung cancer. *Nature Cancer* **1**, 394-409 (2020).
13. Guo G, *et al.* A TNF-JNK-Axl-ERK signaling axis mediates primary resistance to EGFR inhibition in glioblastoma. *Nat Neurosci* **20**, 1074-1084 (2017).
14. Kleczko EK, Heasley LE. Mechanisms of rapid cancer cell reprogramming initiated by targeted receptor tyrosine kinase inhibitors and inherent therapeutic vulnerabilities. *Mol Cancer* **17**, 60 (2018).
15. Niederst MJ, Engelman JA. Bypass mechanisms of resistance to receptor tyrosine kinase inhibition in lung cancer. *Sci Signal* **6**, re6 (2013).
16. Alexander PB, Wang XF. Resistance to receptor tyrosine kinase inhibition in cancer: molecular mechanisms and therapeutic strategies. *Front Med* **9**, 134-138 (2015).
17. Lee HJ, Zhuang G, Cao Y, Du P, Kim HJ, Settleman J. Drug resistance via feedback activation of Stat3 in oncogene-addicted cancer cells. *Cancer Cell* **26**, 207-221 (2014).
18. Lee Y, Wang Y, James M, Jeong JH, You M. Inhibition of IGF1R signaling abrogates resistance to afatinib (BIBW2992) in EGFR T790M mutant lung cancer cells. *Mol Carcinog* **55**, 991-1001 (2016).
19. Kobayashi Y, *et al.* Characterization of EGFR T790M, L792F, and C797S Mutations as Mechanisms of Acquired Resistance to Afatinib in Lung Cancer. *Mol Cancer Ther* **16**, 357-364 (2017).
20. Janjigian YY, *et al.* Dual inhibition of EGFR with afatinib and cetuximab in kinase inhibitor-resistant EGFR-mutant lung cancer with and without T790M mutations. *Cancer Discov* **4**, 1036-1045 (2014).

21. Yoon BW, *et al.* Comparison of T790M Acquisition Between Patients Treated with Afatinib and Gefitinib as First-Line Therapy: Retrospective Propensity Score Matching Analysis. *Transl Oncol* **12**, 852-858 (2019).
22. Wang S, *et al.* Different characteristics and survival in non-small cell lung cancer patients with primary and acquired EGFR T790M mutation. *Int J Cancer* **144**, 2880-2886 (2019).
23. Li D, *et al.* BIBW2992, an irreversible EGFR/HER2 inhibitor highly effective in preclinical lung cancer models. *Oncogene* **27**, 4702-4711 (2008).
24. Yu HA, Arcila ME, Hellmann MD, Kris MG, Ladanyi M, Riely GJ. Poor response to erlotinib in patients with tumors containing baseline EGFR T790M mutations found by routine clinical molecular testing. *Ann Oncol* **25**, 423-428 (2014).
25. Pao W, *et al.* Acquired resistance of lung adenocarcinomas to gefitinib or erlotinib is associated with a second mutation in the EGFR kinase domain. *PLoS Med* **2**, e73 (2005).
26. Gao SP, *et al.* Mutations in the EGFR kinase domain mediate STAT3 activation via IL-6 production in human lung adenocarcinomas. *J Clin Invest* **117**, 3846-3856 (2007).
27. Gazdar AF. Personalized medicine and inhibition of EGFR signaling in lung cancer. *N Engl J Med* **361**, 1018-1020 (2009).
28. Rosell R, Karachaliou N. Large-scale screening for somatic mutations in lung cancer. *Lancet* **387**, 1354-1356 (2016).
29. Hsu PC, Jablons DM, Yang CT, You L. Epidermal Growth Factor Receptor (EGFR) Pathway, Yes-Associated Protein (YAP) and the Regulation of Programmed Death-Ligand 1 (PD-L1) in Non-Small Cell Lung Cancer (NSCLC). *Int J Mol Sci* **20**, (2019).
30. Flaherty KT, Wargo JA, Bivona TG. YAP in MAPK pathway targeted therapy resistance. *Cell Cycle* **14**, 1765-1766 (2015).
31. You B, *et al.* Inhibition of ERK1/2 down-regulates the Hippo/YAP signaling pathway in human NSCLC cells. *Oncotarget* **6**, 4357-4368 (2015).
32. Hsu PC, *et al.* YAP promotes erlotinib resistance in human non-small cell lung cancer cells. *Oncotarget* **7**, 51922-51933 (2016).
33. Gong K, *et al.* TNF-driven adaptive response mediates resistance to EGFR inhibition in lung cancer. *J Clin Invest* **128**, 2500-2518 (2018).

34. Moll HP, *et al.* Afatinib restrains K-RAS-driven lung tumorigenesis. *Sci Transl Med* **10**, (2018).
35. Kruspig B, *et al.* The ERBB network facilitates KRAS-driven lung tumorigenesis. *Sci Transl Med* **10**, (2018).
36. Hayreh SS, Zimmerman B. Management of giant cell arteritis. Our 27-year clinical study: new light on old controversies. *Ophthalmologica* **217**, 239-259 (2003).
37. Nair AB, Jacob S. A simple practice guide for dose conversion between animals and human. *J Basic Clin Pharm* **7**, 27-31 (2016).
38. Perez-Ruiz E, *et al.* Prophylactic TNF blockade uncouples efficacy and toxicity in dual CTLA-4 and PD-1 immunotherapy. *Nature* **569**, 428-432 (2019).
39. Bertrand F, *et al.* TNFalpha blockade overcomes resistance to anti-PD-1 in experimental melanoma. *Nat Commun* **8**, 2256 (2017).
40. Verheijden RJ, *et al.* Association of Anti-TNF with Decreased Survival in Steroid Refractory Ipilimumab and Anti-PD1-Treated Patients in the Dutch Melanoma Treatment Registry. *Clin Cancer Res* **26**, 2268-2274 (2020).

REVIEWER COMMENTS

Reviewer #1 (Remarks to the Author):

The authors adequately revised manuscript. I would like to add only a few comments below.

- Regarding Comment 3 of Reviewer 1, please provide accession number before release to the public (release date can be set right after publication).

- Regarding Comment 6 of Reviewer 1, the latter answer comment sounds not scientific excuse. The revised data (added densitometry data) is OK. Please carefully reconsider whether the comment is adequate or not.

- Regarding Comment 7 of Reviewer 1, please clearly describe the explanation that "Prednisolone is the active metabolite of prednisone, and unify the description including abbreviation.

- Regarding Comment 12 of Reviewer 1, why STAT3 data was not added in Fig 3E, F, I ?

Reviewer #2 (Remarks to the Author):

In response for T790M, authors stated that "a number of studies have shown that the second generation EGFR TKI afatinib is effective in inhibiting the EGFR T790M", and showed references 18-23. This is partially true, because afatinib is well recognized to have activity against EGFR-T790M only at high concentrations which are generally not feasible in patients. While the combined therapy of afatinib and cetuximab shown in reference 20 was proposed in 2014, it has not been unfortunately approved. To directly show the clinically relevant efficacy of EGFR TKI plus prednisone, osimertinib (an approved third generation EGFR TKI), but not afatinib, should be used for experiments with H1975 cells.

In addition, this reviewer felt that the references used for this reply were inappropriate.

For instance, Ref 21 (Yoon BW et al, Transl Oncol 12,852-858, 2019) reported the "Comparison of T790M acquisition between patients treated with afatinib and gefitinib as first-line therapy as a retrospective study. Yoon et al stated that "Disappointingly, afatinib did not show any statistically significant difference compared to gefitinib in terms of cumulative ratio of T790M and median time to T790M acquisition".

Ref 22 (Wang S et al, Int J Cancer 144, 2880-2886, 2019) reported the efficacy of osimertinib against lung cancer with primary EGFR-T790M or acquired EGFR-T790M mutation. In the paper by Wang et al, description relating afatinib was only one sentence "Several patients benefited from gefitinib or afatinib again after osimertinib progression" in the Discussion section.

Reviewer #3 (Remarks to the Author):

I am glad to see that Gong et al. have made their manuscript more succinct. I am satisfied that my comments have been well received and addressed appropriately in the revised manuscript. Well done!

Reviewer #4 (Remarks to the Author):

The study by Gong et al. discovered an advantageous and persisting effect of adding synthetic glucocorticoids (prednisolone) to EGFR inhibitors, which outperformed combinations with single-acting TNF inhibitors (etanercept) or the more broadly acting TNF inhibitor thalidomide. The study addresses an important clinical problem, and has a timely character in view of many studies that have questioned the role of glucocorticoids in other settings of cancer therapy, moving from beneficial to detrimental. The amount of data is impressive and esp. the in vivo results are very

convincing. The study also includes patient data.

The authors have explained well how the adaptive response may spark ectopic signals beyond the (expected) effects of the inhibited signaling pathway (here RTK), and which may form (perhaps even novel) entry points towards lethality. I found this concept important as it helped to position the novelty aspect of this study. Glucocorticoids are indeed broadly acting and whereas many effects could be regarded as "expected", it was clear that prednisolone on its own had (surprisingly) little effect.

I found that most of the Reviewers' detailed and easy-to-address comments were effectively addressed by the authors, concerning methodology and the clarifications were helpful.

Major comment.

I agree with Reviewer 3 on the impact these findings will have for the clinic, esp. with regard to the role of glucocorticoids. I also concur with Rev 3, still, that the volume of data takes the attention somewhat away from the key messages. Sadly, that problem remained.

Rev 3 expressed concerns about the side effects of glucocorticoids. Actually, I worry less about the typical ones that affect quality of life in non-cancer settings in the long run (hyperglycemia, osteoporosis, psychosis...), but rather for the emergence of GC resistance over time. GR is clearly important but not at all monitored. Have authors checked GR levels throughout the treatment? Glucocorticoids are known to downregulate GR over time. Does this happen here as well or not? Studying this could give clues about the mechanism as well. In this respect, I concur with Reviewer 1 that underlying molecular mechanisms are still missing from the manuscript. Despite the extra explanation in the rebuttal which was helpful, the study stayed somewhat observational (mind you, with spectacular results), but, it remains an open question how prednisone in concert with EGFR inhibition exactly circumvents or blocks the bypass RTK signaling pathway.

Prednisone works via GR, a transcription factor, and given the important role of glucocorticoids in this study, it is surprising a GR-viewpoint is completely missing. Nowhere throughout the main study is a piece of data on GR to be found, eg. transcripts, protein levels, phosphorylated residues,... indicating its level, its activity, or how its activity may have changed in combination with the TKI under study.

Minor comments.

I found the explanation on the doses translating to human usage still difficult to follow. If a dose is given as mg per kg, so mpk, this should be similar across species, right? I am unsure where the 0,4mg/kg (mpk) and 0,08 mpk come from or what I may be missing here (multiple dosages per day?)?

The Figure legends contain also descriptions of results which is rather unconventional.

P7 line 148, the results described here do not support the data, the results in the Figure legends are more accurately described.

P9. Line 204-205. I cannot really agree on the statement about the insulin receptor or IGFR-1. I know what the authors want to say and that is that not all receptors are regulated in the same direction, but this sentence should be rephrased because what is being said now is a clear overinterpretation.

REVIEWER COMMENTS

Reviewer #1 (Remarks to the Author):

Comment: The authors adequately revised manuscript. I would like to add only a few comments below.

Response: We appreciate Reviewer 1's critique. The issues raised have helped us to improve the paper and we are happy to address the remaining issues as outlined below.

Comment- Regarding Comment 3 of Reviewer 1, please provide accession number before release to the public (release date can be set right after publication).

Response: SUB9461010/PRJNA763241. Release upon acceptance of the manuscript.

Comment- Regarding Comment 6 of Reviewer 1, the latter answer comment sounds not scientific excuse. The revised data (added densitometry data) is OK. Please carefully reconsider whether the comment is adequate or not.

Response: As pointed out by Reviewer 1, our newly provided densitometry data is sufficient and adequate to clearly address the question raised in Comment 6. We acknowledge that the comment referred to by Reviewer 1 is not adequate and is redundant and is not used in the paper.

Comment- Regarding Comment 7 of Reviewer 1, please clearly describe the explanation that "Prednisolone is the active metabolite of prednisone, and unify the description including abbreviation.

Response: In NCI Thesaurus (NCIt), it says " Prednisone is a synthetic glucocorticoid drug that is particularly effective as an immunosuppressant, and affects virtually all of the immune system. Prednisone is a prodrug that is converted by the liver into prednisolone (a beta-hydroxy group instead of the oxo group at position 11), which is the active drug and also a steroid. It has a role as a prodrug, an anti-inflammatory drug, an antineoplastic agent, an immunosuppressive agent and an adrenergic agent. It is a 20-oxo steroid, an 11-oxo steroid, a 21-hydroxy steroid, a 17alpha-hydroxy steroid, a glucocorticoid, a 3-oxo-Delta(1),Delta(4)-steroid, a primary alpha-hydroxy ketone and a tertiary alpha-hydroxy ketone."

The explanation was revised with more details: "Prednisone is a synthetic glucocorticoid prodrug that is converted by the liver into prednisolone (a beta-hydroxy group instead of the oxo group at position 11), which is the active steroid." The abbreviation PRDN represents both prednisone and prednisolone, since they both work finally in the form of prednisolone.

Comment- Regarding Comment 12 of Reviewer 1, why STAT3 data was not added in Fig 3E, F, I ?

Response: STAT3 data have now been added to Fig. 3E, F, I. Our data indicate that STAT3 is activated in some EGFR mutant cell lines (H3255, PC9) but not in others (HCC827), and this STAT3 activation is suppressed by prednisolone. A previous study also reported that STAT3 is

not activated in response to EGFR inhibition in HCC827 cells¹. We do not detect STAT3 activation in response to EGFR inhibition in EGFRwt expressing A549 or H441 cells. On the other hand, YAP is activated upon EGFR inhibition in all cell lines and this activation is suppressed by prednisolone. These data indicate that there are some differences in the compensatory signals induced by EGFR inhibition in various cell lines. The key point is that prednisone suppresses the STAT3 activation in the cell lines in which STAT3 is activated by EGFR inhibition.

Reviewer #2 (Remarks to the Author):

Comment: In response for T790M, authors stated that “a number of studies have shown that the second generation EGFR TKI afatinib is effective in inhibiting the EGFR T790M”, and showed references 18-23. This is partially true, because afatinib is well recognized to have activity against EGFR-T790M only at high concentrations which are generally not feasible in patients. While the combined therapy of afatinib and cetuximab shown in reference 20 was proposed in 2014, it has not been unfortunately approved. To directly show the clinically relevant efficacy of EGFR TKI plus prednisone, osimertinib (an approved third generation EGFR TKI), but not afatinib, should be used for experiments with H1975 cells.

In addition, this reviewer felt that the references used for this reply were inappropriate. For instance, Ref 21 (Yoon BW et al, Transl Oncol 12,852-858, 2019) reported the “Comparison of T790M acquisition between patients treated with afatinib and gefitinib as first-line therapy as a retrospective study. Yoon et al stated that “Disappointingly, afatinib did not show any statistically significant difference compared to gefitinib in terms of cumulative ratio of T790M and median time to T790M acquisition”.

Ref 22 (Wang S et al, Int J Cancer 144, 2880-2886, 2019) reported the efficacy of osimertinib against lung cancer with primary EGFR-T790M or acquired EGFR-T790M mutation. In the paper by Wang et al, description relating afatinib was only one sentence “Several patients benefited from gefitinib or afatinib again after osimertinib progression” in the Discussion section.

Response: We thank the reviewer for this important comment. The point is well taken. The inappropriate references have been removed. We have now done all the H1975 experiments (including the animal experiment) with osimertinib. We have replaced the afatinib data for H1975 with osimertinib data. The data are shown in the following figures:

Fig. 5H, H1975 cells: cell viability assay with osimertinib and prednisone.

Fig. 5I, H1975 xenograft tumors, treated with osimertinib and prednisone.

Supplementary Fig. 13, H1975 treatment with osimertinib and prednisone and effects on bypass RTK signaling.

Reviewer #3 (Remarks to the Author):

I am glad to see that Gong et al. have made their manuscript more succinct. I am satisfied that my comments have been well received and addressed appropriately in the revised manuscript. Well done!

Response: Thank you!

Reviewer #4 (Remarks to the Author):

Overall Comments: The study by Gong et al. discovered an advantageous and persisting effect of adding synthetic glucocorticoids (prednisolone) to EGFR inhibitors, which outperformed combinations with single-acting TNF inhibitors (etanercept) or the more broadly acting TNF inhibitor thalidomide. The study addresses an important clinical problem, and has a timely character in view of many studies that have questioned the role of glucocorticoids in other settings of cancer therapy, moving from beneficial to detrimental. The amount of data is impressive and esp. the in vivo results are very convincing. The study also includes patient data.

The authors have explained well how the adaptive response may spark ectopic signals beyond the (expected) effects of the inhibited signaling pathway (here RTK), and which may form (perhaps even novel) entry points towards lethality. I found this concept important as it helped to position the novelty aspect of this study. Glucocorticoids are indeed broadly acting and whereas many effects could be regarded as “expected”, it was clear that prednisolone on its own had (surprisingly) little effect.

I found that most of the Reviewers' detailed and easy-to-address comments were effectively addressed by the authors, concerning methodology and the clarifications were helpful.

Response: We appreciate the positive comments and have attempted to address all the concerns.

Major comment.

Comment: I agree with Reviewer 3 on the impact these findings will have for the clinic, esp. with regard to the role of glucocorticoids. I also concur with Rev 3, still, that the volume of data takes the attention somewhat away from the key messages. Sadly, that problem remained.

Response: We acknowledge that there is a substantial amount of data in the manuscript. Based on Reviewer 3 comments (as part of the first revision), we did consolidate a number of figures and removed some data.

Comment: Rev 3 expressed concerns about the side effects of glucocorticoids. Actually, I worry less about the typical ones that affect quality of life in non-cancer settings in the long run (hyperglycemia, osteoporosis, psychosis....), but rather for the emergence of GC resistance over time. GR is clearly important but not at all monitored. Have authors checked GR levels throughout the treatment? Glucocorticoids are known to downregulate GR over time. Does this happen here as well or not? Studying this could give clues about the mechanism as well. In this respect, I concur with Reviewer 1 that underlying molecular mechanisms are still missing from the manuscript. Despite the extra explanation in the rebuttal which was helpful, the study stayed somewhat observational (mind you, with spectacular results), but, it remains an open question how prednisone in concert with EGFR inhibition exactly circumvents or blocks the bypass RTK signaling pathway.

Prednisone works via GR, a transcription factor, and given the important role of glucocorticoids in this study, it is surprising a GR-viewpoint is completely missing. Nowhere throughout the main study is a piece of data on GR to be found, eg. transcripts, protein levels, phosphorylated residues,... indicating its level, its activity, or how its activity may have changed in combination with the TKI under study.

Response: This is an important question. Secondary resistance is a major problem in most cancer therapies. In our animal experiments the combination of EGFR inhibition plus prednisone remained effective for the duration of the experiments (up to 6 months, Figure 4). We did not see a waning of efficacy that would suggest the emergence of secondary resistance.

We undertook the following new experiments focused on the glucocorticoid receptor. The data are shown in Supplementary Fig. 9.

1. GR levels were examined following treatment of lung cancer cell lines with EGFR inhibitor, prednisolone or a combination for 24 hours (Supplementary Fig. 9A). We did not detect a significant change in GR level.
2. The same experiment was conducted after treatment with drugs for 7 days (Supplementary Figure 9B). Although a downregulation of the GR in these lines were detected, GR continues to be present, consistent with the persistent biological effects of glucocorticoids.
3. As expected, we see a substantial increase in phosphorylation of the GR in response to glucocorticoids, but not to erlotinib. The phosphorylation of the GR with the combination treatment is similar to prednisolone alone (Supplementary Fig. 9A).
4. We also examined GR levels in resected tumor tissue after one month of treatment with glucocorticoid. Similar as in vitro experiment for 7 days, the expression level of GR decreases partially (Supplementary Fig. 9C).
5. We also examined the transcriptional activity of the GR using a GRE luciferase reporter. As expected prednisolone induced a strong response, while erlotinib alone has no effect and the combination of prednisolone and erlotinib is similar to prednisolone alone (Fig. 9D-E).

The GR downregulation is expected since stimulation of a receptor by its ligand would be expected to result in downregulation of the receptor. We did detect downregulation of GR after long time treatment of glucocorticoid in multiple NSCLC cell lines and xenografts. However, the remaining GR still presents and may contribute the prolonged efficacy of combined treatment with EGFR inhibition and glucocorticoids. The partial downregulation of the GR is also consistent with long term efficacy of glucocorticoids in other diseases.

Minor comments.

1. I found the explanation on the doses translating to human usage still difficult to follow. If a dose is given as mg per kg, so mpk, this should be similar across species, right? I am unsure where the 0,4mg/kg (mpk) and 0,08 mpk come from or what I may be missing here (multiple dosages per day)?

Response: It turns out that the dose is not necessarily similar across species². From the introduction of the cited article (Nair and Jacob):

“ It should be emphasized that the common perception of scaling of dose based on the body weight (mg/kg) alone is not the right approach. This is primarily because the biochemical, functional systems in species vary which in turn alter pharmacokinetics. Therefore, extrapolation of dose from animals to humans needs consideration of body surface area, pharmacokinetics, and physiological time.”²

This article has been cited 2023 times and we have used guidelines in this article to undertake the mouse to human calculation.

2. The Figure legends contain also descriptions of results which is rather unconventional. P7 line 148, the results described here do not support the data, the results in the Figure legends are more accurately described.

Response: The results described were revised as “Although erlotinib-induced genes in the TNF pathway were blocked by the three drugs at different strength, the TNF gene were almost equally blocked by the three inhibitors (Fig. 1C). ”, consistent with in the legends.

3. P9. Line 204-205. I cannot really agree on the statement about the insulin receptor or IGFR-1. I know what the authors want to say and that is that not all receptors are regulated in the same direction, but this sentence should be rephrased because what is being said now is a clear overinterpretation.

Response: We have removed the sentence that was considered to be a clear overinterpretation.

1. Lee HJ, Zhuang G, Cao Y, Du P, Kim HJ, Settleman J. Drug resistance via feedback activation of Stat3 in oncogene-addicted cancer cells. *Cancer Cell* **26**, 207-221 (2014).
2. Nair AB, Jacob S. A simple practice guide for dose conversion between animals and human. *J Basic Clin Pharm* **7**, 27-31 (2016).

REVIEWERS' COMMENTS

Reviewer #1 (Remarks to the Author):

I am satisfied with the re-revised manuscript. I do not have any additional comments.

Reviewer #2 (Remarks to the Author):

Authors revised MS appropriately.

Reviewer #4 (Remarks to the Author):

The study by the team of Habib is impressive, includes important findings to advance the field and esp. the in vivo results are very convincing. All of my comments were addressed in an appropriate, complete and professional manner. I congratulate the authors for making these further improvements.

I applaud also the stamina of the authors to have gone the extra mile to provide experimental insights on the glucocorticoid receptor, which was before a blind spot in the study, but now more than appropriately addressed and discussed.

The only final "downside" if I can call it this way, is that the title makes no reference at all to the use of glucocorticoids in the study which is a pity since some researchers in the basic or applied nuclear receptor field may miss picking up the study. The study is of such relevance that a most broad readership should pick it up. But perhaps this is something that could be discussed and remedied, if agreed, with the editorial team.

NCOMMS-21-07417B

Response to Reviewers

Reviewer #1 (Remarks to the Author):

I am satisfied with the re-revised manuscript. I do not have any additional comments.

Response: Thank you.

Reviewer #2 (Remarks to the Author):

Authors revised MS appropriately.

Response: Thank you.

Reviewer #4 (Remarks to the Author):

The study by the team of Habib is impressive, includes important findings to advance the field and esp. the in vivo results are very convincing. All of my comments were addressed in an appropriate, complete and professional manner. I congratulate the authors for making these further improvements.

I applaud also the stamina of the authors to have gone the extra mile to provide experimental insights on the glucocorticoid receptor, which was before a blind spot in the study, but now more than appropriately addressed and discussed.

The only final "downside" if I can call it this way, is that the title makes no reference at all to the use of glucocorticoids in the study which is a pity since some researchers in the basic or applied nuclear receptor field may miss picking up the study. The study is of such relevance that a most broad readership should pick it up. But perhaps this is something that could be discussed and remedied, if agreed, with the editorial team.

Response: We appreciate the kind comments. Thank you.

We have changed the title of the paper as recommended by Reviewer 4. The new title is "Comprehensive targeting of resistance to inhibition of RTK signaling pathways by using glucocorticoids".